# Artificial local magnetic field inhomogeneity enhances $T_2$ relaxivity

Zijian Zhou[1,2,*], Rui Tian[1,2,*], Zhenyu Wang[3], Zhen Yang[2], Yijing Liu[2], Gang Liu[1], Ruifang Wang[3], Jinhao Gao[4], Jibin Song[2], Liming Nie[1] & Xiaoyuan Chen[2]

Clustering of magnetic nanoparticles (MNPs) is perhaps the most effective, yet intriguing strategy to enhance $T_2$ relaxivity in magnetic resonance imaging (MRI). However, the underlying mechanism is still not fully understood and the attempts to generalize the classic outersphere theory from single particles to clusters have been found to be inadequate. Here we show that clustering of MNPs enhances local field inhomogeneity due to reduced field symmetry, which can be further elevated by artificially involving iron oxide NPs with heterogeneous geometries in terms of size and shape. The $r_2$ values of iron oxide clusters and Landau–Lifshitz–Gilbert simulations confirmed our hypothesis, indicating that solving magnetic field inhomogeneity may become a powerful way to build correlation between magnetization and $T_2$ relaxivity of MNPs, especially magnetic clusters. This study provides a simple yet distinct mechanism to interpret $T_2$ relaxivity of MNPs, which is crucial to the design of high-performance MRI contrast agents.

[1] State Key Laboratory of Molecular Vaccinology and Molecular Diagnostics & Center for Molecular Imaging and Translational Medicine, School of Public Health, Xiamen University, Xiamen 361102, China. [2] Laboratory of Molecular Imaging and Nanomedicine, National Institute of Biomedical Imaging and Bioengineering, National Institutes of Health, Bethesda, Maryland 20892, USA. [3] Department of Electronic Science and Fujian Key Laboratory of Plasma and Magnetic Resonance, Xiamen University, Xiamen 361005, China. [4] State Key Laboratory of Physical Chemistry of Solid Surfaces, The Key Laboratory for Chemical Biology of Fujian Province, and Department of Chemical Biology, College of Chemistry and Chemical Engineering, Xiamen University, Xiamen 361005, China. * These authors contributed equally to this work. Correspondence and requests for materials should be addressed to J.S. (email: jibin.song@nih.gov) or to L.N. (email: nielm@xmu.edu.cn) or to X.C. (email: shawn.chen@nih.gov).

Magnetic nanoparticles (MNPs) have received tremendous attention in a variety of fields such as data storage[1], biochemical analysis[2,3] and biomedicine[4,5]. Magnetic resonance imaging (MRI) is a powerful and non-invasive technique to detect lesions in soft tissue[6]; however, it often suffers from limited contrast that has encouraged the exploitation of MNPs as MRI contrast agents[7–9]. To pursue high-performance MRI contrast agents, the outersphere motional averaging regime (MAR) theory has been applied to guide the rational engineering of MNPs[10,11]. In short, MAR theory illustrates that $T_2$ contrast efficiency ($r_2$) is related to the diffusion of surrounding water protons and the physical properties of MNPs, which can be tuned by their size[12,13], shape[14–18], component[19–21], crystal structure[22,23] and surface property[24,25].

Another intriguing phenomenon is that cluster formation from single MNPs can cause a marked decrease of $T_2$ relaxation times and thus increase of $r_2$ values, or vice versa[26]. On this basis, magnetic resonance switch (MRSw) systems have been developed to detect a wide range of targets, from metal ions to tumour cells[27–29]. The $T_2$ relaxivity of magnetic clusters is dependent on a number of parameters such as size, frame composition and water penetrating behaviour[30,31]. However, the underlying mechanism of the clustering effect is not clear and the attempts to generalize the principles of single-particle relaxivity to clusters have been confronted with a great deal of complications[26]. On clustering, for example, it is assumed that the decreased surface-to-volume ratio may attenuate the effective radius compared with that of isolated single MNPs. Moreover, the long-range order of magnetic spins during the formation of aggregated clusters is considerably preserved at the same level, which deprives the possibility to enhance the magnetization of clusters[32]. Considerable attention has been focused on controlling the self-assembly of MNPs into clusters, while having variable $r_2$ values with large deviations[33–35]. The exploitation of this prevailing effect, especially the potential mechanism behind it, is thus appealing to guide the rational design of magnetic clusters for MRI and MRSw sensing applications.

Diffusion of protons around the local magnetic field induced by MNPs may lead to the loss of proton spin coherence, namely dephasing, which forms the fundamental basis of $T_2$ contrast agents[36,37]. Although manufacturers have taken great efforts to make a uniform magnet, the magnetic field in an MRI machine is still not perfectly uniform. The considerable level of field inhomogeneity gives rise to $T_2^\star$ decay from the intrinsic inhomogeneous magnet. This requires a spin-echo array to refocus the signal because $T_2^\star$ decay is an order-of-magnitude faster than the intrinsic $T_2$ decay by atomic and molecular mechanisms[6]. Encouraged by this rationale, MNPs have been developed as $T_2$ contrast agents due to the ability to induce local field inhomogeneity[38].

It has been partially identified that the saturated magnetization ($M_s$) value is directly proportional to the $T_2$ relaxivity ($r_2$) of MNPs in MAR; however, the prediction of $r_2$ from the $M_s$ value is based on the averaged behaviour of proton dephasing around given MNPs[10,11]. Although the relationship between proton $T_2$ dephasing and $M_s$ value has been well established in single-domain MNPs, it is still not clear how proton dephasing is related to the $M_s$ value of multi-domain magnetic clusters. In addition, the static dephasing regime (SDR)[39] and echo-limiting regime[36] mechanisms take part when the size of MNPs exceeds the threshold of MAR, further complicating the understanding of $T_2$ relaxivity in magnetic clusters. Here we show that the $T_2$ relaxivity of MNPs can be linked to the local magnetic field inhomogeneity induced by MNPs. Moreover, we are able to artificially tune the local magnetic field inhomogeneity of clusters through integrating multicomponent single MNPs with different geometries in terms of size and shape, thus increasing the $T_2$ relaxivity of clusters.

## Results

**Rationale.** Magnetization of MNPs induces local inhomogeneity of magnetic field that is related to the $M_s$ value of MNPs. To gain broader significance, we reason that the $M_s$ value can be solved as induced field inhomogeneity, and that adjacent MNPs may generate higher degrees of local field complexity over single MNPs due to the reduced symmetry of local magnetic field. Therefore, it is assumed that the increased level of field inhomogeneity may largely enhance the perturbation of proton phase coherence as water molecules diffuse around adjacent MNPs (Fig. 1), which may be the direct reason for the enhanced $T_2$ relaxivity of magnetic clusters. In light of the fact that non-spherical MNPs show enhanced $T_2$ relaxivity compared to spherical ones with equivalent solid volume[14,15], it is natural to assume the role of shape anisotropy in inducing local field

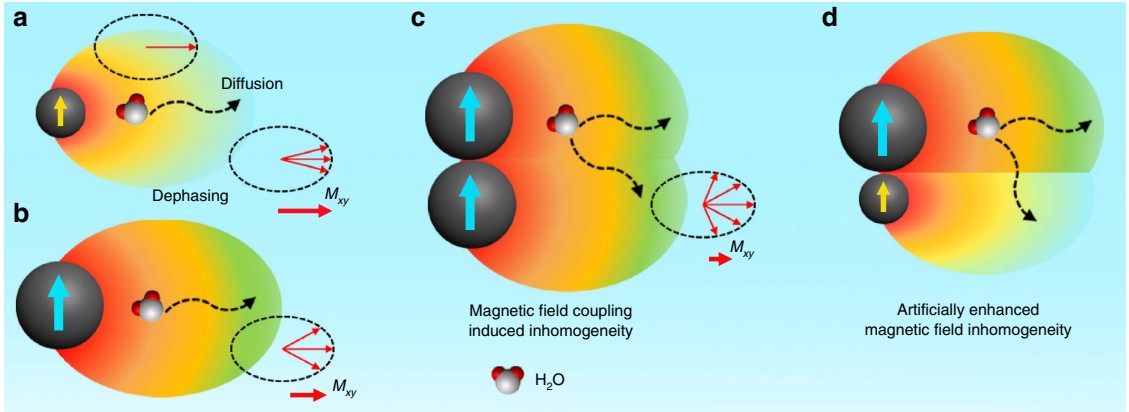

**Figure 1 | Proton diffusion and dephasing around magnetic nanoparticles.** (**a,b**) Magnetic nanoparticles (MNPs) with a larger size or magnetization generate a higher degree of magnetic field gradient, leading to higher level of proton dephasing when water molecules diffuse around MNPs. $M_{xy}$ represents the net nuclei spins on the transverse plane. Dashed arrows indicate the proposed route for water molecule diffusion and dephasing. (**c**) The magnetic field coupling between adjacent MNPs increases the level of field inhomogeneity, which confers greatly enhanced effect to the loss of phase coherence. (**d**) Magnetic field inhomogeneity can be further elevated by artificially involving MNPs of different sizes in proximity, owing to the reduced magnetic field symmetry around MNPs.

inhomogeneity. To further test this assumption, we speculate that field inhomogeneity can be artificially enhanced through assembling different MNPs with heterogeneous local field geometries. We employed iron oxide NPs (IO NPs) with different sizes and shapes as models. The heterogeneity in field geometry may greatly reduce the symmetry and augment the inhomogeneity of the induced local magnetic field, thus conferring a strong influence on the $T_2$ relaxation shortening effect (Fig. 1).

**MRI performance of iron oxide clusters**. We employed two types of IO NPs with average diameters of 5.2 and 15.1 nm as units to explore the clustering effect on $T_2$ relaxivity, denoted as single IO-5 and IO-15 NPs (Supplementary Methods), respectively. By finely controlling the synthetic conditions, we show that both IO-5 and IO-15 are monodispersed with small size deviations of 4.4% and 2.9%, respectively (Supplementary Fig. 1). The IO NPs were pre-decorated with poly(methyl methacrylate) (PMMA) and polyethylene glycol (PEG) to allow for self-assembly through a similar process as reported previously[40]. To obtain IO clusters with different components, IO-5 only, IO-15 only, and IO-5 plus IO-15 were used as building blocks to form IO clusters C1, C2 and C3, respectively. The ratio of IO-5 and IO-15 NPs in IO cluster C3 was set as 1:1 with respect to iron mass. Transmission electron microscopy (TEM) and scanning electron microscopy images show uniform spheres with diameters of $115.5 \pm 10.4$, $127.8 \pm 13.4$ and $129.2 \pm 11.2$ nm for C1, C2 and C3, respectively (Fig. 2a–c; Supplementary Fig. 2). The inter-particle distances ($L$) were $\sim 1.0$ nm as revealed by high-resolution TEM images (Fig. 2a–c). Dynamic light scattering analysis revealed similar hydrodynamic diameters of $134.3 \pm 37.2$, $136.9 \pm 26.8$ and $151.5 \pm 28.1$ nm for C1, C2 and C3, respectively (Fig. 2d). It is noteworthy that the IO clusters are stable in aqueous solution for at least 45 days without obvious agglomeration or change in hydrodynamic diameter (Supplementary Fig. 3), probably due to the PEG decorated surface on the IO clusters. Magnetic hysteresis ($M$–$H$) curves measured at 300 K indicate that both single IO-5 and IO-15 NPs exhibit typical superparamagnetism, but with different saturation magnetization ($M_s$) values of 43.18 and 65.10 e.m.u. g$^{-1}$ of NP weight, respectively (Fig. 2e). Both the size and saturation magnetization are directly proportional to the $T_2$ relaxivity of the IO-5 and IO-15 NPs. On clustering, IO clusters C1–C3 exhibit slightly lower $M_s$ values than their single components, which could be attributed to two factors including the weight proportion of non-magnetized organic polymers in clusters and the demagnetization effect by intra-cluster dipolar interactions[41]. On the other hand, the intra-cluster dipolar interactions by the formation of IO clusters would enhance the dynamic magnetic response of clusters over single NPs[41]. By changing the molecular weight of PMMA, we also obtained IO clusters C4 and C5 with similar components as C3, but with different inter-particle distances $L$ of about 0.1 and 5.0 nm, respectively (Supplementary Fig. 4).

To evaluate the $T_2$ MRI performance, we measured the $r_2$ values of single IO NPs and IO clusters on a 7 T MRI scanner. The $r_2$ values of IO clusters C1 and C2 are $231.6 \pm 9.3$ and $358.3 \pm 14.2$ mM$^{-1}$ s$^{-1}$, respectively (Fig. 2f; Supplementary Table 1). These values are approximately three-fold higher than those of single IO-5 and IO-15 NPs, $70.2 \pm 5.7$ and $127.4 \pm 3.4$ mM$^{-1}$ s$^{-1}$, respectively (**$P < 0.01$). Here we found that the $r_2$ value of C3 is increased to $533.4 \pm 13.2$ mM$^{-1}$ s$^{-1}$ (**$P < 0.01$), significantly higher than those of C1 and C2 (Fig. 2f), whereas simply mixing monodispersed single IO-5 and IO-15 NPs exhibited averaged $T_2$ relaxation times compared

to each single component at different concentrations (Supplementary Fig. 5). We further prepared IO C3$_{low}$ and C3$_{high}$ samples in which the iron mass ratios are 2:1 and 1:2 (that is, the number ratios are 54:1 and 13.5:1) with respect to IO-5 to IO-15 NPs, respectively (Supplementary Fig. 6). The results showed $r_2$ values of $295.8 \pm 23.9$ and $486.6 \pm 18.5$ mM$^{-1}$ s$^{-1}$ for IO C3$_{low}$ and C3$_{high}$ samples, respectively, indicating a vital role of particle distribution and fraction in producing local field inhomogeneity (that is, $T_2$ relaxivity). However, the optimal ratio of IO-5 and IO-15 NPs for obtaining the highest $r_2$ value of IO cluster C3 model needs to be further determined by both experiments and simulations. An MR phantom study confirmed that higher $r_2$ values lead to more prominent $T_2$ contrasts even at low iron concentrations (Fig. 2g). On the other hand, the $r_2$ values of IO clusters C4 and C5 are $515.3 \pm 18.9$ and $445.8 \pm 24.6$ mM$^{-1}$ s$^{-1}$, respectively (Supplementary Table 1). Therefore, the inverse correlation between $r_2$ value and inter-particle distance rules out the influence of water penetration effect on $T_2$ relaxivity[30]. The decrease of the $r_2$ value with increased inter-particle distance in C5 could be due to the reduced magnetic field coupling effect between adjacent MNPs attenuate the local field inhomogeneity and/or the enhanced motion of individual particles that is more likely to average out the proton dephasing. These results indicate that the formation of clusters from IO NPs with different sizes and magnetizations could confer intriguing effects on $T_2$ relaxation enhancement, which is also dependent on the inter-particle distance within clusters.

**Landau–Lifshitz–Gilbert simulations**. To investigate the magnetizing features at the nanoscale, we employed Landau–Lifshitz–Gilbert (LLG) algorithms to calculate the stray field and the stray field gradient induced by MNPs[42–45]. To study the field gradient surrounding the IO NPs, we first calculated the differential of the stray field induced by an IO NP as a function of distance $\left( \Delta H_i = \frac{\partial H}{\partial i}, i = x, y, z \right)$. The results showed that only the components of the stray field gradient in the $x$ and $z$ directions are considered when an external magnetic field is applied along the $+x$ direction (Supplementary Fig. 7).

We then showed that single IO-5 and IO-15 NPs exhibit a magnetization-dependent stray field and field gradient around each particle (Supplementary Fig. 8). In light of the outersphere theory, the $M_s$ value of MNPs is hereby specified as the induced local field inhomogeneity by MNPs, provided that single IO NPs are well dispersed without aggregation. When increasing the particle size, the maximal $r_2$ value is scaled in the SDR where the diffusion distance of protons is negligible compared with the particle size[39]. The $T_2$ relaxivity reaches a plateau and is independent of the size of magnetic particles in the SDR. The onset size threshold of the SDR for IO NPs is in the neighborhood of 30–50 nm, depending on a number of parameters such as the particle characteristics (for example, size and magnetization) and experimental parameters of the pulse sequence[15,26]. We have obtained IO clusters in the condition of the SDR, but with significantly different $r_2$ values, which motivated us to revisit the diffusion and dephasing of protons nearby. To simplify, we simulated the stray field and stray field gradient of models with two IO NPs in line with that of IO clusters C1–C5 (Fig. 3a–f and Supplementary Fig. 9). Since proton dephasing at the inner zone of MNPs is considered too fast to allow for MRI signal acquisition, we set an out-of-plane distance ($d$) of 3 nm as the inner zone and further calculated the magnetization at $d = 5$ nm in our models[36]. First, we showed that magnetic field coupling between IO NPs does not lead to a higher stray field or field gradient because the compact geometry partially neutralizes the magnetic field coupling effect (Supplementary Fig. 10). However,

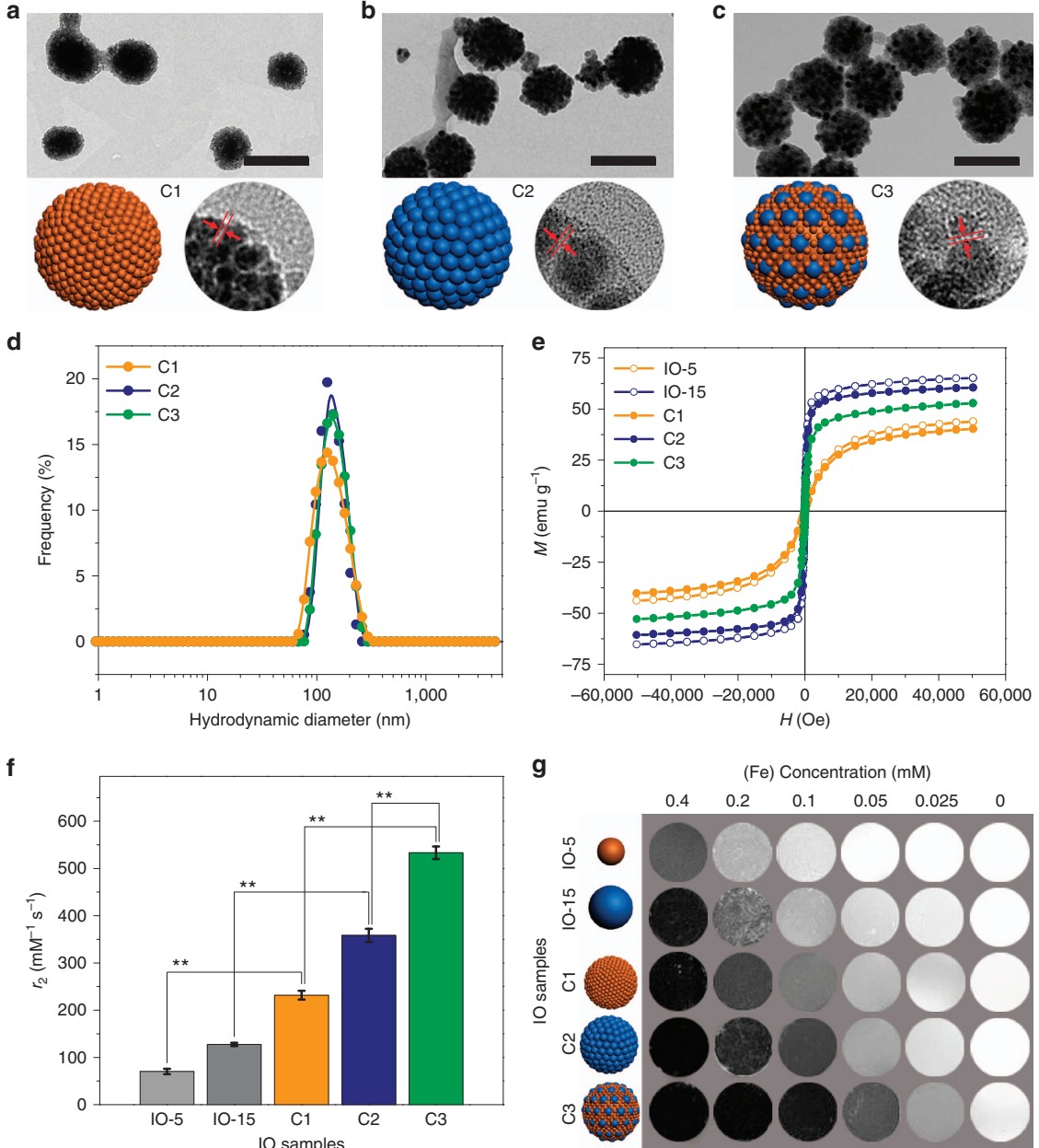

**Figure 2 | Iron oxide clusters C1–C3.** (**a**–**c**) Transmission electron microscopy (TEM) and high-resolution TEM images, as well as cartoons of iron oxide (IO) clusters C1–C3, respectively. Scale bars, 200 nm. The inter-particle distances ($L$) are around 1 nm. (**d**) Dynamic light scattering (DLS) measurements indicate uniform hydrodynamic diameters of IO clusters C1–C3. (**e**) Magnetic hysteresis ($M–H$) curves of single IO-5 and IO-15 NPs, and IO clusters C1–C3 measured at 300 K. (**f**) Columns show $r_2$ values of IO clusters C1–C3, as well as the single IO-5 and IO-15 NPs. Mean values ± s.d.; $n = 3$ (**$P < 0.01$). (**g**) MR phantom of different IO samples show concentration- and relaxivity-dependent contrasts. Darker contrast indicates stronger $T_2$ relaxation enhancement for each concentration. All MRI studies were conducted on a 7 T MRI scanner.

the regularized magnetic field coupling between proximal IO NPs offers a greatly enhanced field inhomogeneity, which could explain the increased $r_2$ values of clusters over those of single particles (Fig. 3g,h and Supplementary Fig. 11). To better represent the field inhomogeneity around the IO clusters, we further conducted LLG simulations of multiple-particle models which imply obvious differences in field asymmetry around the cluster models C1, C2, and C3 (Supplementary Fig. 12). The 3D field inhomogeneity around the cluster models C1, C2, and C3 are visualized in Supplementary Movie 1.

We further resolved a number of models with different model settings (C1, C2, C3, and C3'), such as the relative orientation of adjacent particles to external magnetic field (Supplementary

Figs 13 and 14). The strongest stray field gradient of clusters C3 or C3' is in agreement with the obtained highest $r_2$ value of IO cluster C3. Moreover, these results indicate that the induced local magnetic field inhomogeneity by adjacent particles is greatly dependent on the orientation of the particles relative to the direction of the external magnetic field. The fundamental basis lies in the phenomena of proton diffusion and dephasing around MNPs, which is considerable as an averaged behaviour from a batch of protons (Supplementary Fig. 15). $T_2$ relaxivity is considered independent on the size of particles in SDR, whereas our results further indicate that nanoscopic magnetic field inhomogeneity plays an important role in $T_2$ relaxation enhancement. The abstraction of static stray field strength and

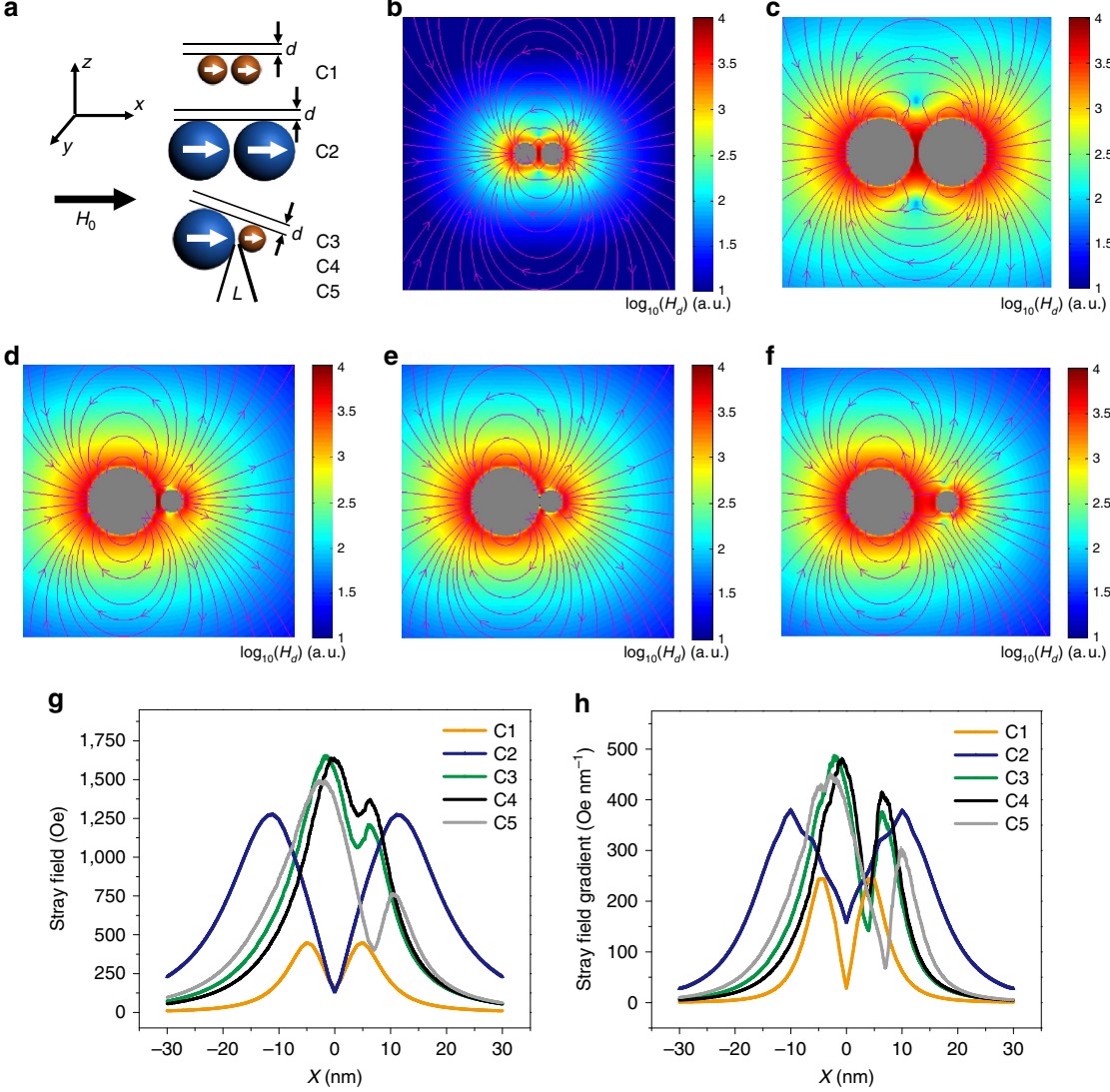

**Figure 3 | Landau–Lifshitz–Gilbert simulations. (a)** Cartoon models representing the simulations of two adjacent iron oxide nanoparticles (IO NPs) of different sizes. The inter-particle distance ($L$) is 1.0 nm in models C1–C3. The models C4 and C5 are derived from model C3 but with different $L$ of 0.1 and 5.0 nm, respectively. **(b–f)** Landau–Lifshitz–Gilbert simulation results of the stray field for model C1–C5, respectively. The $L$ is 1 nm (**b–d**), 0.1 nm (**e**) and 5.0 nm (**f**). Colour bars represent $\log_{10}(H_d)$ (a.u.), where $H_d$ is the calculated stray field. **(g,h)** Calculated stray field and stray field gradient of C1–C5 with an out-of-plane distance ($d$) of 3 nm.

diffusive field gradient may be a way to propagate the classic proton dephasing mechanism in both MAR and SDR.

**Clusters of iron oxide cubes and plates**. We also employed IO cubes and IO plates as examples to study the magnetic field inhomogeneity in $T_2$ relaxation enhancement of anisotropically shaped magnetic nanostructures. The IO cubes with side length of 12 nm and the IO plates with side length of 12 nm and thickness of 4.8 nm have the equivalent solid volume as the IO-15 spheres (Supplementary Fig. 16). Previously, we have shown that anisotropically shaped iron oxide nanostructures exhibited higher $r_2$ values due to the enhanced effective radius, which could generate higher levels of local field inhomogeneity[14,23]. In this study, we artificially combined IO-5 NPs with IO cubes (C6) or plates (C7) as building blocks for self-assembly into clusters.

TEM and HRTEM images show that IO clusters C6 and C7 are of similar diameters of $85.7 \pm 13.9$ and $91.6 \pm 15.3$ nm (Fig. 4a,b). Dynamic light scattering measurements further confirmed the

equivalent hydrodynamic diameters for C6 and C7 (Fig. 4c). LLG simulations based on the models of an adjacent IO-5 and IO cube (C6) or IO-5 and IO plate (C7) were performed under the same conditions as conducted on models C1–C3 (Fig. 4d–f and Supplementary Fig. 17). Because of their non-spherical shape, IO cubes and plates assemble with different orientations with respect to IO-5 NPs. Therefore, we also conducted the LLG simulations of C6 and C7 models of different configurations, C6-bx, C7-bx, C6-bz and C7-bz (Supplementary Figs 18 and 19).

The results are in good agreement with that derived from Fig. 4d–f, indicating the generality of the simulation models. MR relaxivity results showed that IO cubes and plates possess higher $r_2$ values than IO-15 with an equivalent solid volume (Fig. 4g). The highest $r_2$ value of $204.5 \pm 11.4 \, \text{mM}^{-1} \, \text{s}^{-1}$ of IO plates can be attributed to the largest effective radius among IO cubes and IO-15 NPs (Supplementary Table 1). On the contrary, we found that the $r_2$ value of IO cluster C7 is relatively lower than that of C6, $487.7 \pm 21.5$ versus $589.3 \pm 26.8 \, \text{mM}^{-1} \, \text{s}^{-1}$, which may be attributed to the decrease of the averaged effective radius of

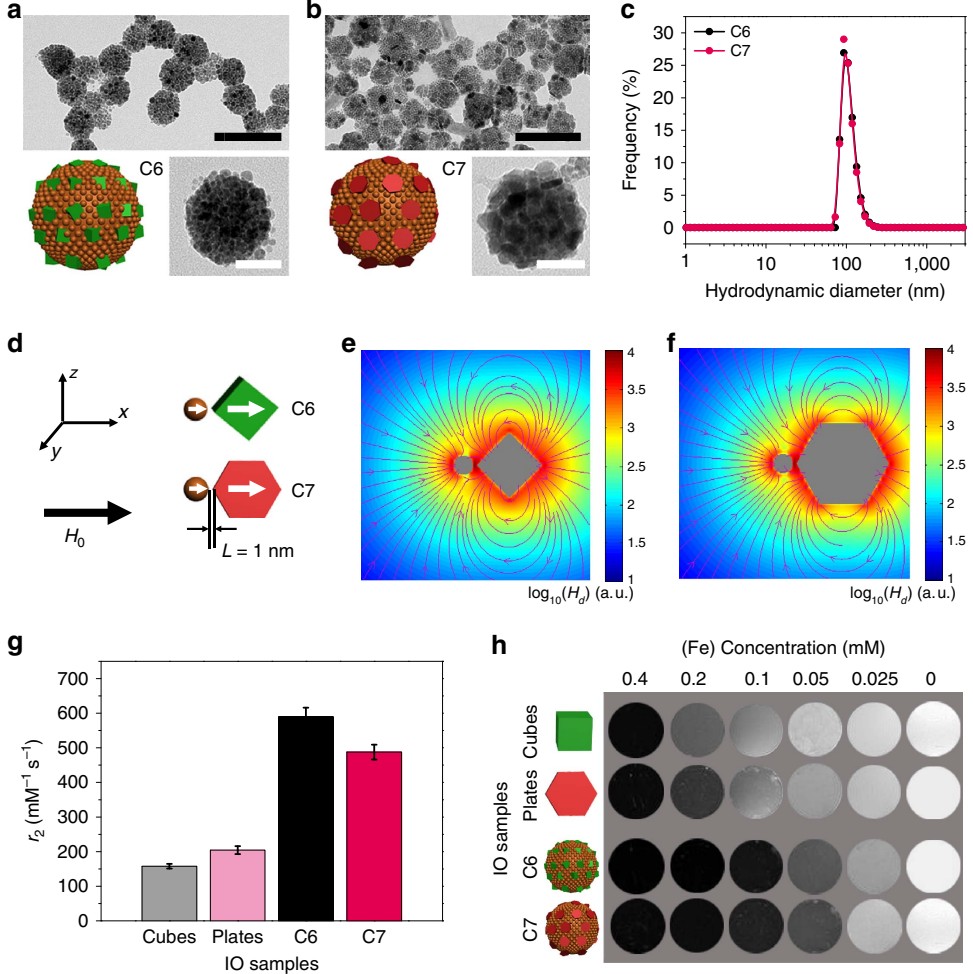

**Figure 4 | Iron oxide clusters C6 and C7.** (**a**,**b**) Transmission electron microscopy (TEM) and high-resolution TEM images, as well as cartoons of iron oxide (IO) clusters C6 and C7, respectively. Scale bars, 200 nm (black), 50 nm (white). (**c**) Dynamic light scattering (DLS) measurements indicate uniform hydrodynamic diameters of IO clusters C6 and C7. (**d**–**f**) Simulation models and the calculated stray fields for the models C6 and C7, respectively. The inter-particle distance $L$ is 1.0 nm. Colour bars represent $\log_{10}(H_d)$ (a.u.), where $H_d$ is the calculated stray field. (**g**) Columns show $r_2$ values of the IO clusters C6 and C7, as well as the single IO cubes and plates. Mean values ± s.d.; $n = 3$. (**h**) MR phantom of different IO samples show concentration- and relaxivity-dependent contrasts. Darker contrast indicates stronger $T_2$ relaxation enhancement for each concentration. All MRI studies were conducted on a 7 T MRI scanner.

nanoplates following Lamer assembly in clusters (Fig. 4g). Prominently, the $r_2$ value of C7 is still higher than that of clusters consisting of only IO plates as reported in previous literature[23], which could be due to the artificially enhanced local field inhomogeneity by introducing IO-5 NPs in close proximity to each other. The highest $r_2$ value of IO cluster C6 is thus attributed to the presence of spin-polarized IO cubes in clusters, which have elevated levels of field inhomogeneity compared to that of IO spheres. The imaging contrasts in MR phantom are consistent with the relaxivities of IO plates and cubes, as well as clusters C6 and C7, at a set of concentrations (Fig. 4h). Therefore, we have demonstrated that artificially enhanced local field inhomogeneity is likely a general strategy to develop high-performance $T_2$ contrast agents based on magnetic clustering structures.

***In vitro* study and *in vivo* MRI of liver tumours.** Prior to conducting *in vivo* MRI studies, we assessed the biocompatibility of IO clusters *in vitro* using HepG2 and macrophage Raw 264.7 cells. After being incubated with IO clusters for 24 h, both HepG2 and Raw 264.7 cells had more than 90% viability at

concentrations up to 100 μg Fe ml$^{-1}$ (Supplementary Fig. 20). A histological study revealed good biocompatibility with no organ abnormality or lesion observed in mice after being treated with IO clusters (2.0 mg Fe kg$^{-1}$ mouse body weight) for 24 h (Supplementary Fig. 21). Quantitative analysis of cellular uptake by inductively coupled plasma optical emission spectroscopy showed that RAW 264.7 cells absorbed higher levels of IO clusters than HepG2 cells (Supplementary Fig. 22), which could be caused by enhanced inflammatory responses in macrophages[46,47]. Due to the probable formation of a protein corona, single IO NPs also revealed aggregated features in endocytotic vesicles of RAW 264.7 cells as shown in cellular TEM images (Supplementary Fig. 23). However, cells absorbing IO clusters show higher $T_2$ contrast and larger $T_2$ relaxation time changes than those absorbing single IO NPs (Supplementary Fig. 24). Clinically used $T_2$ contrast agents (for example, Resovist and Endorem) can exhibit a high degree of uncontrollable particle aggregation and therefore high $r_2$ values (up to 200 mM$^{-1}$ s$^{-1}$) in biological applications[24,48]. Our results suggest that a synthetic approach through well-controlled self-assembly may be a general solution to obtain reliable and efficient $T_2$ contrast agents[34].

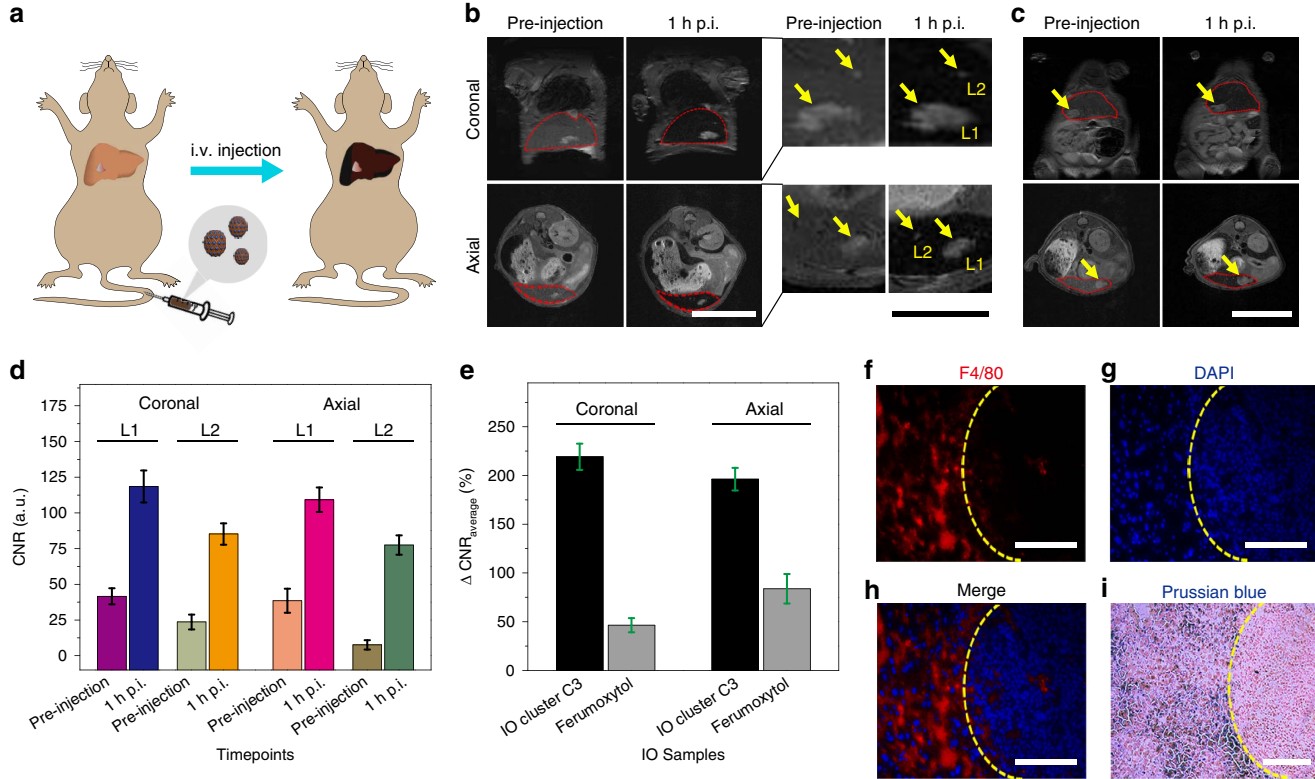

**Figure 5 | Contrast-enhanced MRI of mouse liver tumour.** (**a**) Scheme of contrast-enhanced $T_2$-weighted MRI of an orthotopic mouse liver tumour by intravenous (i.v.) injection of contrast agents. The accumulation of contrast agents in liver tissue resolves liver tumour due to the dark pseudo-contrast found in the liver tissue but not in tumour. (**b,c**) Coronal and axial MR images acquired at pre-injection and 1 h post injection (p.i.) of IO cluster C3 or ferumoxytol at a region of interest (ROI) in the liver (red dotted circle), respectively. The potential lesions in the liver are indicated by yellow arrows. Scale bars, 2 cm (white), 1 cm (black). (**d**) Contrast-to-noise ratio (CNR) analysis of L1 and L2 calculated by fine analysis of MR images at multiple slices using Image J. Both coronal and axial images were analysed on L1 and L2 independently and triply ($n = 3$). Mean values ± s.d. (**e**) Quantitative analysis of $\Delta CNR_{average}$ with respect to imaging liver tumour by IO cluster C3 and commercial ferumoxytol at coronal and axial planes. Mean values ± s.d. (**f–h**) F4/80, DAPI staining and the merged images of the liver tissue, respectively. Yellow dotted curves indicate the boundary between normal liver tissue (left) and the liver tumour (right). Scale bars, 100 μm. (**i**) Prussian blue staining of the liver tissue indicates the decomposition of iron in normal liver tissue but not in the liver tumour. Scale bar, 100 μm.

We then conducted an *in vivo* MRI study in mouse models. Intravenously injected IO clusters are prone to accumulate in the liver through the mononuclear phagocyte system due to the fact that the mouse liver contains a large population of Kupffer cells and other macrophages[49,50]. Therefore, pseudo-contrast imaging is applied to enhance liver tumour contrast, as normal liver tissue show dark $T_2$ signal owing to the accumulation of contrast agents (Fig. 5a). The orthotopic liver tumour model was established by injecting firefly luciferase-encoded HepG2 cells into the mouse liver[51]. Before MRI tests, we used bioluminescence imaging to confirm the presence of liver tumour (Supplementary Fig. 25). A preliminary $T_2$ MRI study indicated that the maximal signal-to-noise (SNR) ratio in the liver was reached around 1 h post injection (p.i.) of IO cluster C3 (Supplementary Fig. 26). Subsequently, we acquired MR images of mice within regions of interest (ROIs) of the liver and the liver tumour both pre-injection and 1 h.p.i. of IO cluster C3, with a dose of 1.0 mg Fe kg$^{-1}$ mouse body weight.

Compared with the surrounding liver tissue, the liver tumour is considerably devoid of Kupffer cells and thus is less likely to uptake contrast agents. Our results show that contrast-enhanced $T_2$ images exhibit a significantly higher degree of recognition for a major liver tumour (L1) with a clear boundary to surroundings, which may greatly improve diagnostic accuracy (Fig. 5b; Supplementary Fig. 27). More interestingly, a small lesion (L2) was clearly seen at 1 h.p.i. due to the suppression of the

background signal, which is barely detectable in pre-injection images (Fig. 5b).

Furthermore, we compared the feasibility of detecting liver tumours with our IO clusters to a clinically approved IO agent, ferumoxytol, with an $r_2$ value of $103.4 \pm 6.5$ mM$^{-1}$ s$^{-1}$ (Supplementary Fig. 28). Under similar conditions, ferumoxytol shows relatively poor $T_2$ contrast and contrast-to-noise ratio (CNR) enhancement in detecting a liver tumour (Fig. 5c; Supplementary Fig. 29). Quantitative analysis of the CNR ratio (CNR = |SNR$_{tumour}$ − SNR$_{liver}$|/SNR$_{tumour}$) of L1 and L2 indicates that the CNR is significantly improved in the 1 h.p.i. images compared with the pre-injection images (Fig. 5d). The average changes of CNR ($\Delta CNR_{average}$ = |CNR$_{post}$ − CNR$_{pre}$|/CNR$_{pre}$) show that IO C3 achieved greatly enhanced $T_2$ contrast efficiency and sensitivity in detecting liver lesions (Fig. 5e). The presence of macrophages in liver tissue but not in liver tumour was confirmed by F4/80 staining (Fig. 5f–h), which is also indicated by the features of iron deposition from Prussian blue staining (Fig. 5i).

## Discussion
Several strategies have been employed to control the uniformity of nanomaterials at different levels, including direct synthesis of single NPs and their self-assembly into superstructures[52–55]. In most cases, homogeneity governs the physiochemical property of materials at the nanoscale and is far beyond aesthetic demands

for materials being functional in nature[56]. For example, the optical emission of quantum dots is highly dependent on the uniformity of structural parameters including size and shape, which are particularly relevant for investigating the direct structure–activity relationship[57,58]. However, the magnetic field generated by MNPs is highly inhomogeneous, where the strength rapidly decreases with increasing distance away from the magnetic center. Based on proton spin–spin relaxation, it has been shown that proton diffusion around MNPs would cause the loss of phase coherence and consequently the shortening of proton $T_2$ relaxation time[36,37]. Apparently, local magnetic fields induced by MNPs is the major cause for $T_2$ relaxation enhancement of protons on MNPs. Although the related parameter $M_s$ has been used to calculate the expected $r_2$ values of given MNPs[10,11], it was still not clear how the inhomogeneous magnetic field impacts the dynamic diffusion and dephasing of protons around MNPs. This becomes more complicated when attempting to generalize the outersphere theory of single-domain particles to multi-domain clusters.

We are specifically interested in the local magnetic field inhomogeneity caused by MNPs and its influence on proton $T_2$ relaxation enhancement. For the first time, we have emphasized that magnetic field inhomogeneity can be resolved into static field strength and diffusive field gradient. We show that adjacent MNPs can enhance the field inhomogeneity due to the reduced symmetry of induced magnetic field by individual particles, which is applicable to explain the higher $T_2$ contrast efficiency of polynuclear clusters over single MNPs. We further envision that proximal MNPs with different magnetizations in terms of size and shape can artificially reduce the symmetry of the induced field, thus enhancing the local field inhomogeneity. The polynuclear IO clusters were made through a well-controlled self-assembly strategy with sizes in the typical SDR range[26]. This allows us to directly compare their $T_2$ relaxivity, focusing on the magnetic field inhomogeneity while isolating the influence of confounding parameters, such as the cluster size and water diffusivity. It is noteworthy that the $r_2$ values of IO clusters would decrease as the size decreases to the size threshold of MAR. Our results demonstrated that the $r_2$ value of IO clusters gradually increases, with increasing local field inhomogeneity, from small to large IO NPs and from spherical to non-spherical IO NPs. The $T_2$ relaxivity of IO clusters is mainly attributed to the outersphere contribution by which the contribution to $T_1$ relaxivity is negligible. On the contrary, small-sized single IO-5 NPs exhibit appreciable $T_1$ contrast enhancement due to the considerable innersphere contribution. More importantly, the high $r_2/r_1$ ratios of these IO clusters further support the great efficacy in $T_2$ contrast imaging (Supplementary Table 1). The higher $r_2$ but lower $M_s$ values of IO cluster C3 relative to that of C2 is a typical example in which local field inhomogeneity is more dependent on the diffusive field gradient rather than field strength.

Previously, Monte Carlo simulations were employed to investigate the relationship between $r_2$ values and geometric parameters of magnetic clusters, indicating that $r_2$ values are sensitive to the changes of particle configuration in cluster, especially in MAR[26]. However, the investigation on the magnetic field of polynuclear models consisting of particles with different geometries (for example, size and shape) is rarely done. In this study, LLG simulations were used to study the magnetization process on nanoscopic magnets, allowing us to elaborate on the magnetic field induced by single MNPs or polynuclear models. The LLG simulations were carried out on well-organized models due to the ease of computational processing and calculation. The symmetry of induced local magnetic field by the MNPs is critical as indicated by the phenomenon of proton diffusion and dephasing. Our results show that IO clusters consisting of IO-5

and IO-15 NPs exhibited higher $r_2$ values over that of clusters consisting of IO-5 or IO-15 NPs only. The highest $r_2$ value was obtained by replacing IO-15 NPs with IO cubes of equivalent solid volume, due to the enhanced level of local field complexity and inhomogeneity. On the other hand, the $r_2$ value was compromised by replacing IO-15 NPs with IO plates, because the Lamer assembly of nanoplates in clusters probably reduces the magnetic field inhomogeneity. For the *in vivo* study, although IO NPs were considered to have low systemic toxicity[59], excessive iron deposition was recently realized to induce extensive cellular damage, particularly fibrosis in the heart and liver[60]. Note that the dose (1.0 mg Fe kg$^{-1}$) of IO clusters used in contrast-enhanced MRI of mouse liver tumour is half the dose commonly used in other studies. This may remarkably reduce the potential of systemic toxicity for pre-clinical research and potential clinical practices.

In summary, this study was devoted to the understanding of how magnetization influences $T_2$ contrast efficiency in magnetic clusters, especially when the classic outersphere theory faces challenges. Using both experiments and theoretical simulations, we have presented that magnetic field inhomogeneity (for example, gradient and symmetry) is the major cause of $T_2$ relaxation enhancement in IO clusters. Artificially enhanced field inhomogeneity boosts the $r_2$ value of IO clusters consisting of IO NPs with different sizes and shapes. LLG calculations further confirmed that architectural heterogeneity with reduced symmetry between MNPs would augment the level of magnetic field inhomogeneity. These simple yet distinct results are amenable to shed light on classic outersphere theory to interpret the $T_2$ shortening effect extended from single particles to magnetic clusters, and more importantly, suggest design considerations of polynuclear clusters for high-performance $T_2$ MRI applications.

## Methods

**Preparation of iron oxide clusters.** Amphiphilic IO NPs were synthesized using the tandem 'grafting to' and 'grafting from' reaction we have recently developed. In the 'grafting to' reaction, a solution of phosphonated-PEG (PEG-P) (20 mg) and the ATRP initiator BiBEP (15 mg) in chloroform (2 ml) was added slowly into 5 ml of the IO NPs (100 nM) in chloroform. After stirring for 12 h, the samples (IO@PEG/BiBEP) were washed with a mixture of acetone and diethyl ether (2:1) and collected after centrifugation at 4,000g for 5 min. IO@PEG/BiBEP was dispersed in 4 ml of DMF for further uses. In the 'grafting from' reaction, MMA (0.4 ml) and IO@PEG/BiBEP (50 nM) were mixed in DMF (2 ml). After being degassed for 30 min by $N_2$, CuBr (6 mg) and ME$_6$TREN (35 mg) were added, and the reaction solution was kept in a water bath at 40 °C for 12 h. After being purified by magnetic separation five times, the IO NPs coated with PEG and PMMA (IO@PEG/PMMA) were stored in chloroform. To prepare IO cluster C3 for example, 60 μl (850 nM) of IO-5@PEG/PMMA and 40 μl (50 nM) of IO-15@PEG/PMMA in chloroform were mixed with 900 μl (0.05 mg ml$^{-1}$) of SDS in water and emulsified for ~5 min by pulsed sonication (100 W and 22.5 kHz, XL-2000 series). The emulsion was then stirred at room temperature for 10 h to evaporate the organic solvent. Other IO cluster samples were prepared by the similar method but with different feed molar ratios and starting IO NPs.

**MR phantom study and relaxivity measurements.** MRI tests were conducted on a Bruker 7 T scanner (Pharmascan) equipped with a small animal-specific body coil. Samples of IO clusters or single IO NPs containing 1% of agarose gel were prepared in tubes with different concentrations of 0.4, 0.2, 0.1, 0.05 and 0.025 mM with respect to iron mass. The MR images were obtained using spin echo sequence with parameters as follows: repetition time = 3,000 ms, echo time = 10, 20, 30, 40, 50, 60, 70, 80, 90, 100, 110, 120, 130, 140, 150, 160 ms, matrix = 256 × 256, field of view = 40 × 40 mm$^2$, slice thickness = 3.00 mm. The $T_2$ relaxation times were calculated by fitting these multiple spin echo images.

**LLG simulation and calculation.** The micromagnetic simulations were carried out by solving the LLG equation:

$$\frac{d\mathbf{M}}{dt} = -\frac{\gamma}{1+\alpha^2} \times \mathbf{M} \times \mathbf{H}_{\text{eff}} - \frac{\gamma\alpha}{(1+\alpha^2)M_s} \times \mathbf{M} \times (\mathbf{M} \times \mathbf{H}_{\text{eff}}) \qquad (1)$$

In this equation, $\vec{H}_{\text{eff}}$ is the effective field and $M_s$ is the saturation magnetization.

We set the gyromagnetic ratio $\gamma = 1.78 \times 10^7\,s^{-1}\,T^{-1}$, the damping constant $\alpha = 1$, the exchange stiffness constant as $1.0e\text{-}11\,J\,m^{-1}$, the Gilbert damping constant as 1.0, and unit cell dimensions of $1 \times 1 \times 1\,nm^3$. The saturation magnetization $M_s$ was determined from the experimental data. The effective field contains contributions from the static external field, the magnetostatic field, the anisotropy field, and the exchange field. The magnetostatic field is calculated after the NP reaches a stable magnetization state under a static external field of 7 T. The stray field gradient is then calculated by the equation:

$$|\nabla \mathbf{H}_d| = \sqrt{(\partial|\mathbf{H}_d|/\partial x)^2 + (\partial|\mathbf{H}_d|/\partial y)^2 + (\partial|\mathbf{H}_d|/\partial z)^2} \quad (2)$$

The magnetic field data from LLG was tabulated with Matlab to show the magnetic field distribution outside the single IO NPs or IO cluster models.

**Mouse orthotopic tumour model.** HepG2 cells were obtained from the American Type Culture Collection (ATCC) and grown in DMEM cell culture medium. Cells were collected with 1 mM EDTA in calcium–magnesium-free phosphate-buffered saline (PBS) and resuspended in serum-free culture medium. Matrigel was used to avoid diffusion out of the injected cells. Prior to injection, the cells ($5 \times 10^5$) were mixed with sterile Matrigel (BD Biosciences) at a 1:1 ratio and kept on ice until injection. The nude mouse was anaesthetized with isoflurane inhalation. The abdomen was sterilized with iodine and alcohol swabs and a small (0.5–1 cm) incision was made in the skin above the liver. The syringe was gently advanced into the liver and the tumour cell mixture was injected. The abdominal wall was sutured using a 6.0 vicryl absorbable suture and a 6.0 prolene nonabsorbable suture for the skin layer.

***In vivo* study of contrast-enhanced $T_2$-weighted MRI.** All animal studies were conducted following a protocol approved by the National Institutes of Health Clinical Center Animal Care and Use Committee (NIH CC/ACUC). Mice were anaesthetized by isoflurane (1.0–2.0%) in oxygen and placed in an animal-specific body coil for MRI data acquisition. Mice were kept warm by circulating warm water (37 °C) and were placed in a stretched prone position with a respiratory sensor during the experiments. $T_2$-weighted images were acquired at pre-injection and post injection (p.i.) of contrast agents intravenously on both coronal and axial planes, focusing on the ROI of mouse liver. Multi-slice multi-echo sequence was employed to acquire images using parameters as follows: repetition time $= 2,000$ ms, echo time $= 30$ ms, flip angle $= 180°$, matrix size $= 256 \times 256$, field of view $= 40 \times 40\,mm^2$, slices $= 16$, slice thickness $= 1$ mm. A MR compatible small animal respiratory gating device was utilized to reduce the artefacts caused by respiration. MR images were analysed by measuring signal intensity in ROI with NIH supported software Image J. SNR was calculated according to the methods provided by the National Electrical Manufacturers Association (NEMA) standards publication (MS 6-2008, R2014) on single-image measurement procedure for SNR. The following equations were used to deduce CNR: $SNR = SI_{mean}/SD_{noise}$; $\Delta SNR = |SNR_{post} - SNR_{pre}|/SNR_{pre}$; $CNR = |SNR_{tumour} - SNR_{liver}|/SNR_{tumour}$; $\Delta CNR = |CNR_{post} - CNR_{pre}|/CNR_{pre}$.

**Data availability.** The authors declare that the data supporting the findings of this study are available within the article and its Supplementary Information Files or from the corresponding author on reasonable request.

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

## Acknowledgements

This work was supported by the National Science Foundation of China (81571744 and 81601489), the National Basic Research Program of China (863 Program 2015AA020502), the Fundamental Research Funds for the Central Universities (20720170065), the Science Foundation of Fujian Province (No. 2014Y2004), and by the Intramural Research Program (IRP), National Institute of Biomedical Imaging and Bioengineering (NIBIB), National Institutes of Health (NIH). We thank Drs O. Jacobson, I. Weiss and G. Niu for helping to initiate orthotopic liver tumours, Drs J. Munasinghe and X. Yang for helping to acquire MRI data, and Dr Henry S. Eden for proof-reading the manuscript.

## Author contributions

Z.Z., X.C. and L.N. conceived and designed the project; Z.Z., R.T., J.S., Z.Y. and Y.L. performed the experiments; Z.W. and R.W. performed the LLG calculations; Z.Z., R.T., J.S., J.G., G.L., L.N. and X.C. analysed the data; and Z.Z., L.N. and X.C. co-wrote the paper. All the authors have discussed the results and have approved the final version.

## Additional information

**Competing interests:** The authors declare no competing financial interests.

