## [Peer Review File · Nature Communications]

Reviewers' comments:

Reviewer #1 (Remarks to the Author):

This manuscript describes a study of the effects of clustering of particles with two different diameters, or particles with different shapes, on the T2 relaxivity and potential MRI contrast properties of iron oxide magnetic nanoparticles. The authors claim to elucidate the effects of clustering and field inhomogeneity on the relaxivity of the particles described. Unfortunately, the manuscript is riddled with mischaracterizations of the literature, claims of novelty where there is none, lack of references to the work of others, and lack of required detail of methods to evaluate rigor. This is not to say the manuscript is completely flawed. The authors do present some interesting results of the relaxivity of clusters and the consequences of using non-spherical particles for MR contrast. However, the following major concerns preclude recommendation for publication:

1. In their introduction the authors appear to attribute to themselves the idea that field inhomogeneity around a magnetic nanoparticle may be responsible for T2 relaxation shortening. The distortion (inhomogeneity) of the local magnetic field caused by a magnetic nanoparticle is well known to be the mechanism by which T2 relaxation shortening of protons occurs. See for example the work of Carroll et al. *Nanotechnology* 21(2010) 035103.
2. The authors are misleading the reader when they state in their introduction that "it is still not clear how proton dephasing is related to Ms value of MNPs". There are well established theories for this. See for example the work of Carroll et al. *Nanotechnology* 21(2010) 035103.
3. In their introduction, the authors appear to claim novelty in the idea that inhomogeneities caused by clusters of magnetic nanoparticles give rise to improved MRI contrast. This has already been established in the work of Riffle and collaborators (*Journal of Materials Chemistry B*, 2013, 1, 1142; *Langmuir*, 2014, 30, 1580).
4. In their introduction, the authors appear to claim that the idea of using non-spherical particles or assemblies is their own and novel. This is simply not the case. The literature has many examples of magnetic nanoparticles of non-spherical shape with "enhanced" MRI contrast properties, such as nanocubes, nanoflowers, nanowhiskers, etc.
5. The nanoparticles used in this study, IO-5 and IO-15 have very different saturation magnetizations. The authors should comment on why this is the case and how this affects their conclusions and comparison with models.
6. Does water penetrate the particles reported here? If so, and if this depends on spacing, then the observed changes in relaxivity could be related to water diffusion into and out of the clusters. This has already been discussed by Pothanyee, *Journal of Materials Chemistry B*, 2013, 1, 1142.
7. The authors refer to the results in Figure S5 as the results of simulations based on the Landau-Lifshitz-Gilbert equation. This cannot be the case. The figure seems to show results for the field distributions surrounding nanoparticles, which is not something one obtains from the LLG equations. The LLG equations describe the change of a dipole in an applied field and are not "field equations", which is what one would need to generate a field distribution.
8. Given the point above, the authors have not provided adequate information to assess the rigor of the methods used to obtain many of the reported results (i.e., Figures 3, 4, S5-S11, S14). Their methods section on the supposed LLG simulations lacks any real detail or references.

9. In page 7 the authors claim they have obtained clusters in the SDR regime (30-50 nm), but most of their clusters are actually much larger (100-200 nm) than the SDR regime (see Figs S2 and S3).

10. The simplistic models of Figures S7 and Figure 3 are not representative of the clusters in Figures 3, S2 and S3. Thus, even if upon revision the authors clarify how their simulations were carried out, their relevance is in doubt.

11. The data in Fig S13 is not enough to prove that the particles in S13b are "plates". The authors should at least provide some form of verification, such as AFM. There is some further evidence for this in Fig 4b, so it might help to refer to that at some point in the manuscript.

12. As noted previously, others (besides the authors) have demonstrated MRI contrast properties of magnetic nanocubes. Their work should be cited.

Reviewer #2 (Remarks to the Author):

The article reported on a new parameter involved in the relaxivity performances of iron oxide nanoparticles : the local magnetic field inhomogeneity which allows enhancing T2 relaxivity. The authors showed that artificial local magnetic field inhomogeneity may be obtained by assembling nanoparticles with different sizes or shapes. They confirmed their results by performing in vitro and in vivo measurements.

Indeed the magnetic properties of the nanoparticles, such as the magnetic moment (μ_c) or the saturation magnetization, the effective or aggregate size (r) and the water diffusion constant around the magnetic core, which is mainly related to the nature of the organic coating are usually reported as key parameters to control in order to design highly contrasting T2 contrast agents. The authors demonstrated thus that the local magnetic field inhomogeneity contributes to the performance of iron oxide contrast agents.

Considering that this parameter is new and allows really understanding some unexplained differences between samples, this paper deserves therefore to be published. However there are the following questions/comments :

- the authors centered mostly the article on the MAR model. However depending on the size of nanoparticles but also on the size of aggregates of nanoparticles, three types of regimes may be observed : MAR, SRD, ELR. As the inhomogeneity is studied in assembled nanoparticles, could the authors discuss these models in the introduction part and be clear on which model fits with the studied samples.

- What would the effect of dipolar interactions induced by the aggregation state of assembled samples ?

- A good contrast agent should display high r_2 value but also a high r_2/r_1 ratio. Could the authors add r_1 values in Table S1 as well as r_2/r_1 ratio ? and make comments ?

indeed T2 contrast agents are mainly based on superparamagnetic nanoparticles, for which the inner-sphere contribution to relaxation is minor as compared with the dominant outer-sphere contribution but iron oxide with small sizes or specific shapes may allow T1 contrast enhancements. This parameter should be considered in the discussion.

- could the authors detail how they perform magnetic measurement ? is the organic part weight contribution removed from saturation magnetization values ?

Reviewer #3 (Remarks to the Author):

Recommendation: Publish in Nature Communications after major revision.

Summary comments: In this manuscript, the authors envisioned that the degree of field inhomogeneity around MNPs is responsible for T2 relaxation shortening effect. They synthesized various kinds of iron oxide clusters, and evaluate their T2 relaxation. They figured out enhanced field inhomogeneity boosts the r_2 value of IO clusters. Also, LLG calculations was used for confirming that architectural heterogeneity with reduced symmetry would augment the level of magnetic field inhomogeneity. This manuscript is well written and suggesting the design considerations of high-performance T2 MRI contrast agents. However, several issues should be addressed before this manuscript become acceptable for Nature Communications.

Our response:

Comment #1: How is the distribution of IO-5 and IO-15? Are they evenly distributed?

Comment #2: In Page 5 line 85, 1:1 iron mass ratio of IO-5 and IO-15 was used for C3. It means that the number ratio is 27:1 and C3 is predominantly composed of IO-5. If such small number fraction of IO-15 could result in twice the r_2 value, then it would be good to investigate other number ratios to see if there are better r_2 value.

Comment #3: How could the hydrodynamic diameters of C1, C2, and C3 be the same? They had different diameters in TEM images. Authors argued that they were prepared by the same method, but only C3 preparation method was described in the method section. Also, how does the size of IO clusters affect the magnetic field inhomogeneity and T2 relaxation?

Comment #4: Inter-particle distance shown in the manuscript does not seem to be accurate. TEM can see only the distance on the cluster surface. Also, 0.1 nm distance is hard to measure with the TEM images shown. C5 clusters are not rigid. How did you calculate the 5 nm inter particle distance? Scale bars are missing in the inset TEM images in Figure 4a, 4b, S3a, S3b.

Comment #5: In Figures and Supplementary Figures, denoting S1 and S2 is confusing. They were not used in main text. We recommend authors to use IO-5 and IO-15.

Comment #6: In Figure 2e, Are the iron masses in S1, S2, C1, C2, C3 the same? Otherwise you cannot compare them.

Comment #7: In Figure 4e and 4f, IO nanoparticles can face to different side of IO cubes and plates (ex) facets, edges, and vertexes. How can LLG simulation be changed in these cases?

Reviewer #4 (Remarks to the Author):

This manuscript claims to elucidate the source of T2 shortening due to clustering of iron oxide nanoparticles (IONPs). The hypothesis is that local field inhomogeneities lead to enhanced dephasing by reduced field symmetry as compared to a single IONP.

In order to demonstrate this, a two-fold approach involving empirical evidence and Landau-Lifshitz-Gilbert modeling was used. While the experimental results are interesting, I find that the LLG modeling is insufficient: rather than model nanoparticle clusters involving many nanoparticles, the authors employ a 2-nanoparticle model and claim that this can be generalized (line 127).

I do not see how some of the modeling shown in supplementary figures S10 and S11 are in good agreement with the obtained r^2 values. There is little rigorous explanation here.

The differences in transverse relaxivity due to geometry or composition of clustered nanoparticles is interesting. I just find that the agreement with modeling is vague. Perhaps it would be useful to focus on these results. It would be interesting to know how stable these NP clusters are in aqueous solution, as colloidal stability has been routinely demonstrated to be a problem in IONP systems.

There is no mention of how CNR is calculated in that the SNR calculation is unspecified. This begs the question of how reproducible the CNR results would be on another scanner, even of the same model and field strength.

I feel that a different journal would be a better place to present this work, with much more detail provided on the LLG modeling, which should be expanded to a much larger number of particles to better represent the actual clusters being studied.

The following are our point-by-point response (Re) to the Reviewer comments (in Italics). Changes in the manuscript have been marked in red.

Reviewer #1 (Remarks to the Author):

[This manuscript describes a study of the effects of clustering of particles with two different diameters, or particles with different shapes, on the T2 relaxivity and potential MRI contrast properties of iron oxide magnetic nanoparticles. The authors claim to elucidate the effects of clustering and field inhomogeneity on the relaxivity of the particles described. Unfortunately, the manuscript is riddled with mischaracterizations of the literature, claims of novelty where there is none, lack of references to the work of others, and lack of required detail of methods to evaluate rigor. This is not to say the manuscript is completely flawed. The authors do present some interesting results of the relaxivity of clusters and the consequences of using non-spherical particles for MR contrast. However, the following major concerns preclude recommendation for publication:]

Re: We are grateful for the reviewer's comments to improve our manuscript. The concept of field inhomogeneity in MRI was firstly introduced to describe the intrinsic defect of the magnet itself on MRI machine which results in T_2^* effect. For magnetic nanoparticles (MNPs) as T_2 contrast agents, it has been recognized that the magnetization (M_s) and size (d) are the main factors of local field inhomogeneity generated by MNPs, as in a typical motional average regime. However, this theory is limited to the systems of single-domain MNPs with different sizes and magnetizations (e.g., Carroll et al. *Nanotechnology* 21(2010) 035103). The application of outersphere theory in predicting r_2 value ($r_2 = (256\pi^2\gamma^2/405) \kappa M_s^2 d^2 / (D(1+L/d))$, see ref. 14) of multi-domain magnetic clusters is still controversial. A clear demonstration or characterization of the magnetization (i.e., local field inhomogeneity) effect to r_2 value by magnetic clusters is rarely reported.

In this study, we not only calculated but also artificially tuned the field inhomogeneity in magnetic clusters, aiming to elucidate the clear role of field inhomogeneity in proton T_2 relaxation shortening effect. Hence, we appreciate the reviewer's concerns and made revisions on literature discussion in the Introduction part and provided detailed calculation methods in Methods part.

[1. In their introduction the authors appear to attribute to themselves the idea that field inhomogeneity around a magnetic nanoparticle may be responsible for T2 relaxation shortening. The distortion (inhomogeneity) of the local magnetic field caused by a magnetic nanoparticle is well known to be the mechanism by which T2 relaxation shortening of protons occurs. See for example the work of Carroll et al. Nanotechnology 21(2010) 035103.]

Re: We agree with the reviewer that the distortion of local magnetic field by a magnetic nanoparticle is well known to be the mechanism of T_2 relaxation shortening effect. However, this statement was established based on single-domain nanoparticles but not multi-domain clusters. The provided literature reports the experimental validation of T_2 relaxivity models on single-domain iron oxide nanoparticles with size ranging from 6 to 13 nm. Indeed, the authors found that experimental r_2 values are in good agreement with the theoretical models. However, the field

inhomogeneity revealed by the size and magnetization of single-domain iron oxide nanoparticles is not generalizable to that of multi-domain magnetic clusters as discussed in our study. We feel that our statements may have misled the reviewer. To avoid confusion, we have made appropriate revision and discussed the literature (ref. 37) in the Introduction part accordingly.

Original:

“Encouraged by this rationale, we envision herein that the degree of field inhomogeneity around MNPs may be responsible for conferring the T_2 relaxation shortening effect.”

Revision:

“Encouraged by this rationale, MNPs have been developed as T_2 contrast agents due to the ability to induce local field inhomogeneity³⁷. We further envision herein that the degree of field inhomogeneity around MNPs is directly related to their T_2 relaxivity.”

Additional (References):

“37. Matthew, R.J.C. et al. Experimental validation of proton transverse relaxivity models for superparamagnetic nanoparticle MRI contrast agents. *Nanotechnology* **21**, 035103 (2010).”

*[2. The authors are misleading the reader when they state in their introduction that “it is still not clear how proton dephasing is related to M_s value of MNPs”. There are well established theories for this. See for example the work of Carroll et al. *Nanotechnology* 21(2010) 035103.]*

Re: We agree with the reviewer that the theory relating to the relationship between proton T_2 dephasing and M_s value has been well established on MNPs. The higher M_s value would lead to stronger proton T_2 dephasing effect (i.e., r_2 value). However, the current theory is mainly based on single-domain MNPs. The generalization of this theory from single-domain MNPs to multi-domain magnetic clusters is not in place. We have made revision in the Introduction part accordingly.

Original:

“It is still not clear how proton dephasing is related to M_s value of MNPs and magnetic clusters.”

Revision:

“Although the relationship between proton T_2 dephasing and M_s value has been well established on single-domain MNPs, it is still not clear how proton dephasing is related to M_s value of multi-domain magnetic clusters.”

*[3. In their introduction, the authors appear to claim novelty in the idea that inhomogeneities caused by clusters of magnetic nanoparticles give rise to improved MRI contrast. This has already been established in the work of Riffle and collaborators (*Journal of Materials Chemistry B*, 2013, 1, 1142; *Langmuir*, 2014, 30, 1580).]*

Re: We thank the reviewer’s comments and references. These two papers by Riffle and collaborators mainly focused on the establishment of structure-property relationship between copolymer networks and T_2 relaxivity of magnetic clusters, where the field inhomogeneity induced by magnetic clusters was not discussed. For example, the first paper (*Journal of*

Materials Chemistry B, 2013, 1, 1142) mainly studied the influence of hydrophilic spacing of polymer networks and the size of clustered particles on MR relaxivities. The second paper (*Langmuir*, 2014, 30, 1580) reported a set of analytical theory models to accurately predict T_2 relaxivity of magnetic clusters with tailored compositions and size distributions. With great respect, however, both papers failed to provide a clear demonstration or characterization of the magnetization (i.e., local field inhomogeneity) induced by magnetic clusters. These two papers provided by the reviewer are helpful in understanding T_2 relaxivity on magnetic clusters. We have discussed in the Introduction part and updated the References accordingly.

Additional:

“The T_2 relaxivity of magnetic clusters is relative to a lot of parameters such as size, frame composition, and water penetrating behavior^{29,30}.”

Additional (References):

“29. Pothayee, N. et al. Magnetic nanoclusters with hydrophilic spacing for dual drug delivery and sensitive magnetic resonance imaging. *J. Mater. Chem. B* **1**, 1142-1149 (2013).

30. Balasubramaniam, S. et al. Toward Design of Magnetic Nanoparticle Clusters Stabilized by Biocompatible Diblock Copolymers for T2-Weighted MRI Contrast. *Langmuir* **30**, 1580-1587 (2014).”

[4. In their introduction, the authors appear to claim that the idea of using non-spherical particles or assemblies is their own and novel. This is simply not the case. The literature has many examples of magnetic nanoparticles of non-spherical shape with “enhanced” MRI contrast properties, such as nanocubes, nanoflowers, nanowhiskers, etc.]

Re: Indeed, non-spherical single magnetic nanoparticles have been reported with enhanced T_2 relaxivity compared with the spherical ones with equivalent solid volume. Also, magnetic clusters consisting of non-spherical shaped single-component particle assembly have been published in a great number of literature. However, magnetic clusters consisting of different sized and shaped multi-component MNPs within one cluster was not reported. We have made appropriate revision in the Introduction part and discussed the related references (Page 4) accordingly.

Additional:

“In light of the fact that non-spherical MNPs showed enhanced T_2 relaxivity compared with the spherical ones with equivalent solid volume^{14,15}, the role of shape anisotropy in inducing local field inhomogeneity is considerable.”

[5. The nanoparticles used in this study, IO-5 and IO-15 have very different saturation magnetizations. The authors should comment on why this is the case and how this affects their conclusions and comparison with models.]

Re: We thank the reviewer’s suggestion. In this work, we studied the effect of artificially introduced local magnetic field inhomogeneity on T_2 relaxivity of clusters through assembling IO

NPs with different sizes and shapes. The saturation magnetization of IO NPs is a direct reflection of the long-range-order of magnetic spins in a given nano-entity, which is dependent on their crystallization, temperature (i.e., thermal agitation), and size. Larger sized IO-15 NPs are considered to have higher level of long-range-order of magnetic spins (i.e., higher saturation magnetization value) than smaller sized IO-5 NPs. Both the size and saturation magnetization are directly proportional to the induced local field inhomogeneity of IO NPs. Therefore, this case does not affect our main conclusions in this work. Indeed, IO NPs with same size but different saturation magnetizations are good candidates for our study. However, there are major technical challenges to synthesize such IO NPs. Following the reviewer's suggestion, we have added the discussion in the manuscript (Page 6) accordingly.

Additional:

“Both the size and saturation magnetization are directly proportional to the induced local field inhomogeneity of IO NPs, which allow us to coordinate their effect to T_2 relaxivity of IO-5 and IO-15 NPs.”

[6. Does water penetrate the particles reported here? If so, and if this depends on spacing, then the observed changes in relaxivity could be related to water diffusion into and out of the clusters. This has already been discussed by Pothanyee, Journal of Materials Chemistry B, 2013, 1, 1142.]

Re: The clusters in our study were formed with hydrophobic PMMA at inner and hydrophilic PEG at outer space of the clusters. Regarding the reviewer's concern, we have found literature which reports the possibility for water penetrating into the hydrophobic PMMA network (*J. Phys. Chem. B*, 2009, 113, 13269). However, we observed in our results that increasing the inter-particle distance leads to a decrease of r_2 value when comparing the r_2 values of IO clusters C3-C5. Therefore, water penetration effect may have negligible influence to T_2 relaxivity in our study. This is different from the case studied in the literature provided by the reviewer, in which they employed hydrophilic polymers to construct clusters. In that way, water penetration into the hydrophilic space of clusters increases the T_2 relaxivity (*Journal of Materials Chemistry B*, 2013, 1, 1142). We have revised our discussion accordingly (Page 7).

Additional:

“Therefore, the inverse relationship between r_2 value and inter-particle distance rules out the influence of water penetration effect to T_2 relaxivity²⁹.”

[7. The authors refer to the results in Figure S5 as the results of simulations based on the Landau-Lifshitz-Gilbert equation. This cannot be the case. The figure seems to show results for the field distributions surrounding nanoparticles, which is not something one obtains from the LLG equations. The LLG equations describe the change of a dipole in an applied field and are not “field equations”, which is what one would need to generate a field distribution.]

Re: We agree with the reviewer that LLG equations are initially developed to calculate the change of a dipole in an applied magnetic field. It is also known that LLG equations are able to

map the magnetic field distribution of a nanoparticle system (ref. 34, *Phys. Rev. Lett.* 2013, **110**, 117201). In the Supplementary Fig. 7, we show that only the components of the stray field gradient at x and z directions are considered when an external magnetic field is applied along the +x direction. To this end, we have calculated the differential of the stray field induced by IO NP as a function of distance ($\Delta H_i = \frac{\partial H}{\partial i}$, $i = x, y, z$). This data is a preliminary statement to simplify the related discussion on LLG models. To address the reviewer's concern, we have made appropriate revisions to clarify the statement in the text (Page 7) accordingly.

Original:

“When an external magnetic field is applied along the +x direction, only the components of the stray field and field gradient at x and z directions are considered (**Supplementary Fig. S5**).”

Revision:

“To study the field gradient surrounding IO NPs, we firstly calculated the differential of the stray field induced by IO NP as a function of distance ($\Delta H_i = \frac{\partial H}{\partial i}$, $i = x, y, z$). The results show that only the components of the stray field gradient at x and z directions are considered when an external magnetic field is applied along the +x direction (**Supplementary Fig. 7**).”

[8. Given the point above, the authors have not provided adequate information to assess the rigor of the methods used to obtain many of the reported results (i.e., Figures 3, 4, S5-S11, S14). Their methods section on the supposed LLG simulations lacks any real detail or references.]

Re: Following the reviewer's suggestion, we have provided detailed calculation methods and updated the references.

Additional (Methods):

“The micromagnetic simulations were carried out by solving the Landau-Lifshitz-Gilbert (LLG) equation (Supplementary Refs. 4-6):

$$\frac{d\vec{M}}{dt} = -\frac{\gamma}{1+\alpha^2} \times \vec{M} \times \vec{H}_{eff} - \frac{\gamma\alpha}{(1+\alpha^2)M_s} \times \vec{M} \times (\vec{M} \times \vec{H}_{eff})$$

In this equation, \vec{H}_{eff} is the effective field and M_s is the saturation magnetization. We set the gyromagnetic ratio $\gamma = 1.78 \times 10^7$, the damping constant $\alpha = 1$, the exchange stiffness constant as $1.0e^{-6}$ erg/cm, the Gilbert damping constant as 1.0, and unit cell dimensions of $1 \times 1 \times 1$ nm³. The saturation magnetization M_s was determined from the experimental data. The effective field contains contributions from the static external field, the magnetostatic field, the anisotropy field, and the exchange field. The magnetostatic field was calculated after the nanoparticle reached a stable magnetization state under a static external field of 7 T. The stray field gradient was then calculated by the equation:

$$|\nabla H_d| = \sqrt{(\partial|\vec{H}_d|/\partial x)^2 + (\partial|\vec{H}_d|/\partial y)^2 + (\partial|\vec{H}_d|/\partial z)^2}$$

The magnetic field data from LLG was tabulated with Matlab to show the magnetic field distribution outside the single IO NPs or IO cluster models.”

Additional (Supplementary References):

“4. Cimrak, I. A survey on the numerics and computations for the Landau-Lifshitz equation of micromagnetism. *Arch. Comput. Method Eng.* **15**, 277-309 (2008).

5. Scheinfein, M. R. et al. Micromagnetics of domain-walls at surfaces. *Phys. Rev. B* **43**, 3395-3422 (1991).

6. Scheinfein, M. R. et al. LLG micromagnetics simulator, software for micromagnetic simulations. <http://llgmicro.home.mindspring.com> (1997).”

[9. In page 7 the authors claim they have obtained clusters in the SDR regime (30-50 nm), but most of their clusters are actually much larger (100-200 nm) than the SDR regime (see Figs S2 and S3).]

Re: The statement of 30-50 nm was used to describe the typical onset size threshold for SDR, not the size range for SDR. Smaller sized MNPs are subject to motional average regime (MAR), where the T_2 relaxivity increases with increasing MNP size. When the size reaches the onset threshold of SDR, the T_2 relaxivity reaches a plateau and is independent of the size of magnetic particles. In general, the size threshold of SDR depends on a number of parameters such as the particle characteristics (e.g., size and magnetization) and experimental parameters of the pulse sequence (ref. 25). We have made appropriate revision in the revised manuscript (Page 8) accordingly.

Original:

“The critical size reaching SDR for IO NPs is around 30-50 nm, depending on the geometry and constituents of IO NPs¹⁵.”

Revision:

“The \$T_2\$ relaxivity reaches a plateau and is independent of the size of magnetic particles in SDR. The onset size threshold of SDR for IO NPs is in the neighborhood of 30-50 nm, depending on a set of parameters such as the particle characteristics (e.g., size and magnetization) and experimental parameters of the pulse sequence^{15,25}.”

[10. The simplistic models of Figures S7 and Figure 3 are not representative of the clusters in Figures 3, S2 and S3. Thus, even if upon revision the authors clarify how their simulations were carried out, their relevance is in doubt.]

Re: We thank the reviewer’s comments. The simplistic two-nanoparticle models were carried out due to the ease of understanding and calculation. Regarding the reviewer’s concern, we further conducted LLG modeling of multiple-particle assembly models to better represent the field inhomogeneity around IO clusters. We show that the local field inhomogeneity induced by multiple-particle models imply obvious differences in field asymmetry around IO clusters C1, C2, and C3 in a more comprehensive manner than the two-nanoparticle models. Moreover, we also provided a movie to view in 3D the field inhomogeneity induced by assembled cluster models C1, C2, and C3 (Supplementary Movie 1). These results are in good agreement with the

current results shown in Figures 2 and 3. Accordingly, we have updated the data and discussed in the revised manuscript (Supplementary Fig. 12 and Movie 1, page 9).

Additional:

“To better represent the field inhomogeneity around IO clusters, we further conducted LLG simulation of multiple-particle models which imply obvious differences in field asymmetry around the cluster models C1, C2, and C3 (Supplementary Fig. 12). Moreover, we also provided a movie to view in 3D the field inhomogeneity around the cluster models C1, C2, and C3 (Supplementary Movie 1).”

“Supplementary Fig. 12 LLG simulation results of multiple-particle models with the external magnetic field along +x direction. The stray field (upper row) and stray field gradient (lower row) of the IO cluster models C1, C2, and C3 show obvious differences in field inhomogeneity around the IO cluster models, which are in good agreement with that derived from the two-particle models.”

“Supplementary Movie 1 3D-view of the filed inhomogeneity induced by IO cluster models C1, C2, and C3 with an external magnetic field along +x direction. The movie is a side view of stray field gradient of the spinning multiple-particle models C1, C2, and C3.”

[11. The data in Fig S13 is not enough to prove that the particles in S13b are “plates”. The authors should at least provide some form of verification, such as AFM. There is some further evidence for this in Fig 4b, so it might help to refer to that at some point in the manuscript.]

Re: We thank the reviewer’s comment. Accordingly, we have added an inset TEM image to show vertical nanoplates (now Supplementary Fig. 16b, inset).

[12. As noted previously, others (besides the authors) have demonstrated MRI contrast properties of magnetic nanocubes. Their work should be cited.]

Re: We have updated the related literature (refs. 16-18) in addition to the existing literature (refs. 14, 15) in the revised manuscript.

Additional (References):

“16. Sharma, V.K., Alipour, A., Soran-Erdem, Z., Aykut, Z.G. & Demir, H.V. Highly monodisperse low-magnetization magnetite nanocubes as simultaneous T1-T2 MRI contrast agents. *Nanoscale* **7**, 10519-10526 (2015).

17. Wetterskog, E., Tai, C.-W., Grins, J., Bergström, L. & Salazar-Alvarez, G. Anomalous Magnetic Properties of Nanoparticles Arising from Defect Structures: Topotaxial Oxidation of Fe_{1-x}O|Fe_{3-δ}O₄ Core|Shell Nanocubes to Single-Phase Particles. *ACS Nano* **7**, 7132-7144 (2013).

18. Voros, E. et al. TPA Immobilization on Iron Oxide Nanocubes and Localized Magnetic Hyperthermia Accelerate Blood Clot Lysis. *Adv. Funct. Mater.* **25**, 1709-1718 (2015).”

Reviewer #2 (Remarks to the Author):

[The article reported on a new parameter involved in the relaxivity performances of iron oxide nanoparticles : the local magnetic field inhomogeneity which allows enhancing T2 relaxivity. The authors showed that artificial local magnetic field inhomogeneity may be obtained by assembling nanoparticles with different sizes or shapes. They confirmed their results by performing in vitro and in vivo measurements.

Indeed the magnetic properties of the nanoparticles, such as the magnetic moment (μ) or the saturation magnetization, the effective or aggregate size (r) and the water diffusion constant around the magnetic core, which is mainly related to the nature of the organic coating are usually reported as key parameters to control in order to design highly contrasting T2 contrast agents. The authors demonstrated thus that the local magnetic field inhomogeneity contributes to the performance of iron oxide contrast agents.

Considering that this parameter is new and allows really understanding some unexplained differences between samples, this paper deserves therefore to be published. However there are the following questions/comments:]

Re: We appreciate the reviewer’s careful reading and insightful comments.

[- the authors centered mostly the article on the MAR model. However depending on the size of nanoparticles but also on the size of aggregates of nanoparticles, three types of regimes may be observed: MAR, SRD, ELR. As the inhomogeneity is studied in assembled nanoparticles, could the authors discuss these models in the introduction part and be clear on which model fits with the studied samples.]

Re: MAR model is the most widely applied model to study T_2 relaxivities of MNPs in the literature, which covers single-domain MNPs with size below 30-50 nm. Beyond the size threshold of MAR, SDR describes a plateau of maximal r_2 value which is independent of the size of magnetic particles, while ELR describes the decrease in r_2 value upon further increasing the size of magnetic particles. However, the size threshold for MAR and SDR vary in different cases

which depends on a number of parameters such as the particle characteristics (e.g., size and magnetization) and experimental parameters of the pulse sequence. In our study, IO clusters are about 100 nm in diameter, which fall into the typical SDR. Following the reviewer's suggestion, we have added the discussion in the revised manuscript (Pages 4 and 8) accordingly.

Additional:

“Furthermore, the static dephasing regime (SDR) and echo-limiting regime (ELR) mechanisms take part when the size of MNPs exceeds the threshold of MAR, complicating the understanding of T_2 relaxivity in magnetic clusters.”

“The T_2 relaxivity reaches a plateau and is independent of the size of magnetic particles in SDR. The onset size threshold of SDR for IO NPs is in the neighborhood of 30-50 nm, depending on a number of parameters such as the particle characteristics (e.g., size and magnetization) and experimental parameters of the pulse sequence^{15, 25}.”

[- What would the effect of dipolar interactions induced by the aggregation state of assembled samples?]

Re: The formation of magnetic clusters would give rise to the intra-cluster dipolar interactions. On the one hand, the dipolar interactions of assembled clusters may enhance the dynamic magnetic response to single MNPs. On the other hand, the intra-cluster dipolar interactions would cause a reduction of M_s value due to the demagnetizing effect (*Phys. Chem. Chem. Phys.*, 2016, 18, 10954-10963), which may partially explain our results of the observed decrease in M_s value upon cluster formation. Accordingly, we have updated the discussion and Reference (ref. 39) on this point in the revised manuscript (Page 6).

Additional:

“Upon clustering, IO clusters C1-C3 exhibited slightly lower M_s values compared with their single components, which could be attributed to two factors: (i) the weight proportion of non-magnetized organic polymers in clusters; (ii) the demagnetizing effect by intra-cluster dipolar interactions³⁹.”

Additional (References):

“39. Ovejero, J.G. et al. Effects of inter- and intra-aggregate magnetic dipolar interactions on the magnetic heating efficiency of iron oxide nanoparticles. *Phys. Chem. Chem. Phys.* **18**, 10954-10963 (2016).”

[- A good contrast agent should display high r_2 value but also a high r_2/r_1 ratio. Could the authors add r_1 values in Table S1 as well as r_2/r_1 ratio? and make comments?]

Re: We agree with the reviewer's insightful comments. Accordingly, we have measured the r_1 values and calculated the r_2/r_1 ratios of the magnetic samples (shown in Supplementary Table 1) and updated the discussion in the revised manuscript (Page 14).

Additional:

“More importantly, the high r_2/r_1 ratios of these IO clusters further imply the great potential in T_2 contrast imaging (Supplementary Table 1).”

[Indeed T_2 contrast agents are mainly based on superparamagnetic nanoparticles, for which the inner-sphere contribution to relaxation is minor as compared with the dominant outer-sphere contribution but iron oxide with small sizes or specific shapes may allow T_1 contrast enhancements. This parameter should be considered in the discussion.]

Re: We agree with the reviewer’s insightful comments and discussed this point accordingly (Page 14).

Additional:

“The T_2 relaxivity of IO clusters is mainly attributed to the outersphere contribution by which the contribution to T_1 relaxivity is minimal. On the contrary, small-sized single IO-5 NPs exhibit applicable T_1 contrast enhancement due to the considerable innersphere contribution.”

[- could the authors detail how they perform magnetic measurement? is the organic part weight contribution removed from saturation magnetization values?]

Re: The magnetic measurements were performed on SQUID instrument using dry powder samples which were obtained by sequential solvent washing (3 times) and metal bath drying (100 °C). The organic parts, oleic acid (as-prepared single IO NPs) or PMMA and PEG polymers (IO cluster samples), are not likely to be fully removed because of the mild treatment. The reason for mild treatment in our experiments is to prevent the potential of re-crystallization of IO NPs at high temperature. The larger weight proportion of the organic part leads to lower contribution to magnetization, which decreases the saturation magnetization (M_s) in the IO cluster samples. However, the weight of organic part is a relatively small portion with respect to the majority of inorganic nanoparticles, which has little impact on the overall results. The observed decrease in M_s value of IO clusters C1-C3 compared with that of single components can be attributed to two factors: (i) the weight proportion of non-magnetized organic polymers in clusters; (ii) the demagnetizing effect by intra-cluster dipolar interactions (ref. 39). To address the reviewer’s concern, we have emphasized the experimental details in the Methods part and discussed in the text (Page 6, see also the response to the reviewer’s second comment).

Additional:

“Upon clustering, IO clusters C1-C3 exhibited slightly lower M_s values compared with their single components, which could be attributed to two factors: (i) the weight proportion of non-magnetized organic polymers in clusters; (ii) the demagnetization effect by intra-cluster dipolar interactions³⁹. The larger weight proportion of the organic part leads to lower contribution to magnetization, which decreases the saturation magnetization (M_s) as in IO cluster samples.”

Additional (Methods):

“The preparation of samples for magnetization measurements followed by sequential solvent washing (3 times) and metal bath drying (100 °C) to obtain dry powder samples.”

Reviewer #3 (Remarks to the Author):

[Recommendation: Publish in Nature Communications after major revision.]

[Summary comments: In this manuscript, the authors envisioned that the degree of field inhomogeneity around MNPs is responsible for T₂ relaxation shortening effect. They synthesized various kinds of iron oxide clusters, and evaluate their T₂ relaxation. They figured out enhanced field inhomogeneity boosts the r₂ value of IO clusters. Also, LLG calculations was used for confirming that architectural heterogeneity with reduced symmetry would augment the level of magnetic field inhomogeneity. This manuscript is well written and suggesting the design considerations of high-performance T₂ MRI contrast agents. However, several issues should be addressed before this manuscript become acceptable for Nature Communications.]

Re: We appreciate the reviewer's positive comments.

[Comment #1: How is the distribution of IO-5 and IO-15? Are they evenly distributed?]

Re: The IO-5 and IO-15 NPs in IO clusters (e.g., C3) are randomly distributed according to the observation of high magnified TEM images (Fig. 2c). However, some small IO-5 NPs are likely to occupy the interspace between large IO-15 NPs due to the size mismatch of these two IO NPs.

[Comment #2: In Page 5 line 85, 1:1 iron mass ratio of IO-5 and IO-15 was used for C3. It means that the number ratio is 27:1 and C3 is predominantly composed of IO-5. If such small number fraction of IO-15 could result in twice the r₂ value, then it would be good to investigate other number ratios to see if there are better r₂ value.]

Re: The review is correct that the iron mass ratio of 1:1 for IO-5 and IO-15 NPs is not necessarily the optimal ratio. Following the reviewer's suggestion, we have prepared IO C₃_{low} and IO C₃_{high} samples in which the iron mass ratios are 2:1 and 1:2 with respect to IO-5 to IO-15 NPs, respectively. In other words, the number ratios are 54:1 and 13.5:1 with respect to IO-5 to IO-15 NPs in IO C₃_{low} and IO C₃_{high} samples, respectively. The results showed r₂ values of 295.8 ± 23.9 and 486.6 ± 18.5 for IO C₃_{low} and IO C₃_{high} samples, respectively, indicating a vital role of particle distribution and fraction in producing local field inhomogeneity (i.e., T₂ relaxivity). The optimal ratio of IO-5 and IO-15 NPs for obtaining the highest r₂ value of IO cluster C3 model needs to be further determined by both experimental and simulation results. Following the reviewer's suggestion, we have added these data in Supplementary Fig. 6 and discussed in the text (Page 7) accordingly.

Additional:

“We further prepared IO C₃_{low} and C₃_{high} samples with iron mass ratios of 2:1 and 1:2 (i.e., the number ratios are 54:1 and 13.5:1) with respect to IO-5 to IO-15 NPs (Supplementary Fig. 6).

The results showed r_2 values of 295.8 ± 23.9 and 486.6 ± 18.5 for IO $C3_{low}$ and $C3_{high}$ samples, respectively, indicating a vital role of particle distribution and fraction in producing local field inhomogeneity (i.e., T_2 relaxivity). However, the optimal ratio of IO-5 and IO-15 NPs for obtaining the highest r_2 value of IO cluster C3 model needs to be further determined by both experimental and simulation results.”

Supplementary Fig. 6 (a, b) TEM images of IO cluster $C3_{low}$ and $C3_{high}$ samples. (c) The iron mass ratios are 2:1, 1:1, and 1:2 with respect to IO-5 to IO-15 NPs in IO cluster $C3_{low}$, C3, and $C3_{high}$ samples, while the number ratios are 54:1, 27:1, and 13.5:1, respectively. (d) Comparison of r_2 values for IO cluster $C3_{low}$, C3, and $C3_{high}$ samples ($n = 3$; $**P < 0.01$).”

[Comment #3: How could the hydrodynamic diameters of C1, C2, and C3 be the same? They had different diameters in TEM images. Authors argued that they were prepared by the same method, but only C3 preparation method was described in the method section. Also, how does the size of IO clusters affect the magnetic field inhomogeneity and T_2 relaxation?]

Re: The inorganic core sizes of C1, C2, and C3 are 115.5 ± 10.4 , 127.8 ± 13.4 , and 129.2 ± 11.2 nm, respectively, measured from the TEM images. The hydrodynamic diameters obtained from dynamic light scattering (DLS) measurements are 134.3 ± 37.2 , 136.9 ± 26.8 , and 151.5 ± 28.1 nm for C1, C2, and C3, respectively (Fig. 2d). The hydrodynamic diameters of C1-C3 are not exactly the same, however, the size distribution curves with large deviations of hydrodynamic diameters may have misled the reviewer (Fig. 2d). We have updated these numbers of hydrodynamic diameters for C1-C3 in the text (Page 5).

We claimed that the size of IO clusters C1-C3 falls into a typical SDR regime where the r_2 value is independent of the size of IO clusters. However, as the size of IO clusters further decreases, MAR would take over, in which the r_2 value changes with the size (see also the

response to Reviewer #1). Instead of investigating size effect, we studied the magnetization effect (i.e., local field inhomogeneity) of magnetic clusters to their T_2 relaxivity in our study. Following the reviewer's suggestion, we have added the discussion in the revised manuscript accordingly (Page 14).

In addition, we have updated the Methods part to provide details of preparing other IO clusters besides C3. All the clusters were prepared through a similar method but with different starting IO NPs.

Original:

“Dynamic light scattering (DLS) analysis revealed equivalent hydrodynamic diameters for all the three IO clusters (**Fig. 2d**).”

Revision:

“Dynamic light scattering (DLS) analysis revealed similar hydrodynamic diameters of 134.3 ± 37.2 , 136.9 ± 26.8 , and 151.5 ± 28.1 nm for C1, C2, and C3, respectively (**Fig. 2d**).”

Additional:

“It is noteworthy that as the size of IO clusters decreases to be smaller than 30-50 nm, MAR would take over in which the r_2 value varies with size.”

Additional (Methods):

“Other IO cluster samples were prepared by the similar method but with different feed molar ratios and starting IO NPs.”

[Comment #4: Inter-particle distance shown in the manuscript does not seem to be accurate. TEM can see only the distance on the cluster surface. Also, 0.1 nm distance is hard to measure with the TEM images shown. C5 clusters are not rigid. How did you calculate the 5 nm inter particle distance? Scale bars are missing in the inset TEM images in Figure 4a, 4b, S3a, S3b.]

Re: Indeed, it is difficult to directly measure the particle distances within clusters. During the preparation of IO clusters, we firstly modified single IO NPs with PMMA and PEG polymers on their surface. Therefore, we assume that the inter-particle distance between nanoparticles is generalizable from cluster surface to that of the overall cluster. The IO clusters C4 and C5 with inter-particle distances of 0.1 and 5.0 nm were prepared by using PMMA and PEG polymers with different molecular weights (i.e., chain lengths). The numbers of 0.1 and 5.0 nm were obtained from average values by measuring multiple inter-particle distances on the surface of IO cluster projection TEM images. Additionally, we have updated the scale bars in the inset TEM images in Fig. 4a,b and Supplementary Fig. 4a2, b2 according to the reviewer's comments.

[Comment #5: In Figures and Supplementary Figures, denoting S1 and S2 is confusing. They were not used in main text. We recommend authors to use IO-5 and IO-15.]

Re: We thank the reviewer's comments and have revised in the text and Figures (Fig. 2e, and Supplementary Fig. S8, S10, and S14) accordingly.

[Comment #6: In Figure 2e, Are the iron masses in S1, S2, C1, C2, C3 the same? Otherwise you cannot compare them.]

Re: We prepared the powder samples for magnetization measurements by sequential solvent washing (3 times) and metal bath drying (100 °C). The sample weight is considered from both inorganic and organic residues after the treatment. However, the weight of organic part is a relatively small portion as compared to the majority of inorganic nanoparticles, which would have little effect on the overall results. The organic part led to slightly lower saturation magnetization (M_s) of IO clusters over IO particles. The observed decrease in M_s value of IO clusters C1-C3 compared with that of single components can be attributed to two factors: (i) the weight proportion of non-magnetized organic polymers in clusters; (ii) the demagnetizing effect by intra-cluster dipolar interactions (ref. 39). We have emphasized this point in the revised manuscript accordingly (Page 6, see also the response to the Reviewer #2).

Additional:

“Upon clustering, IO clusters C1-C3 exhibited slightly lower M_s values as compared to their single component, which could be attributed to two factors: (i) the weight proportion of non-magnetized organic polymers in clusters; (ii) the demagnetizing effect by intra-cluster dipolar interactions³⁹.”

[Comment #7: In Figure 4e and 4f, IO nanoparticles can face to different side of IO cubes and plates (ex) facets, edges, and vertexes. How can LLG simulation be changed in these cases?]

Re: We agree with the reviewer’s comments. Accordingly, we conducted additional LLG simulation based on different orientations of IO cubes and IO plates relative to IO-5 NPs. The conclusion is in good agreement with that derived from Fig. 4d-f, indicating the generality of the simulation models. The data has been added in Supplementary Figs. 18 and 19 and discussed in the revised manuscript (Page 10).

Additional:

“Because of the non-spherical shape of IO cube and plate, they opt to assemble with different orientations with respect to IO-5 NPs or external magnetic field. Therefore, we also conducted the LLG simulation of C6 and C7 models of different configurations, C6-bx, C7-bx, C6-bz, and C7-bz (Supplementary Figs. 18 and 19). The results are in good agreement with that derived from Fig. 4d-f, indicating the generality of the simulation models.”

Supplementary Fig. 18 LLG simulations of C6 and C7 with different orientations. (a) Cartoon shows the models C6-bx and C7-bx for calculating the stray field and field gradient with an external magnetic field (7 T) applied along +x direction. (b, c) Simulated stray field for C6-bx and C7-bx, respectively. (d, e) Simulated stray field gradient for C6-bx and C7-bx, respectively. (f, g) Calculated stray field of models C6-bx and C7-bx at $d = 3$ and 5 nm, respectively. (h, i) Calculated stray field gradient of models C6-bx and C7-bx at $d = 3$ and 5 nm, respectively.”

Supplementary Fig. 19 LLG simulations of C6 and C7 with different orientations. (a) Cartoon shows the models C6-bz and C7-bz for calculating the stray field and field gradient with an external magnetic field (7 T) applied along +x direction. (b, c) Simulated stray field for C6-bz and C7-bz, respectively. (d, e) Simulated stray field gradient for C6-bz and C7-bz, respectively. (f, g) Calculated stray field of models C6-bz and C7-bz at $d = 3$ and 5 nm, respectively. (h, i) Calculated stray field gradient of models C6-bz and C7-bz at $d = 3$ and 5 nm, respectively.”

Reviewer #4 (Remarks to the Author):

[This manuscript claims to elucidate the source of T2 shortening due to clustering of iron oxide nanoparticles (IONPs). The hypothesis is that local field inhomogeneities lead to enhanced dephasing by reduced field symmetry as compared to a single IONP.

In order to demonstrate this, a two-fold approach involving empirical evidence and Landau-Lifshitz-Gilbert modeling was used. While the experimental results are interesting, I find that the LLG modeling is insufficient: rather than model nanoparticle clusters involving many nanoparticles, the authors employ a 2-nanoparticle model and claim that this can be generalized (line 127).]

Re: We appreciate the reviewer's insightful comments. We also agree with the reviewer that the LLG modeling of a 2-nanoparticle model is not sufficient to reveal the 3D distribution of magnetic field inhomogeneity induced by IO clusters. To address the reviewer's concerns, we have conducted additional LLG calculations and presented a 3D-view movie of the field inhomogeneity around the assembled cluster models (Supplementary Fig. 12 and Movie 1, see also the response to the Reviewer #1).

[I do not see how some of the modeling shown in supplementary figures S10 and S11 are in good agreement with the obtained r_2 values. There is little rigorous explanation here.]

Re: In Supplementary Figs. 10 and 11 (now Supplementary Figs. 13 and 14 in the revised version), we have shown the LLG simulation results of IO cluster models (C1, C2, C3, and C3') with an external magnetic field at +z direction. This data is supplement to the models shown in Fig. 3 with an external magnetic field at +x direction. Both models C3 and C3' are relative to IO cluster C3 but with different model settings. The two sets of models showed that either model C3 or C3' exhibit the strongest stray field gradient among others, respectively, which are in agreement with the obtained highest r_2 value of IO cluster C3. More importantly, these results further indicate that the induced local magnetic field inhomogeneity by adjacent particles is greatly dependent on the orientation of particles relative to the direction of external magnetic field. Following the reviewer's suggestion, we have updated the discussion on the Supplementary Figs. 13 and 14 accordingly (Page 9).

Original:

“For purposes of comparison, we resolved a number of models with different orientations, showing that field inhomogeneity generated by magnetic field coupling differs from different orientations (**Supplementary Figs. S10 and S11**).”

Revision:

“We further resolved a number of models with different model settings, i.e., relative orientation of adjacent particles to external magnetic field (**Supplementary Figs. 13 and 14**). The strongest stray field gradient C3 or C3' among others is in agreement with the obtained highest r_2 value of IO cluster C3. Moreover, these results indicate that the induced local magnetic field inhomogeneity by adjacent particles is greatly dependent on the orientation of particles relative to the direction of external magnetic field.”

[The differences in transverse relaxivity due to geometry or composition of clustered nanoparticles is interesting. I just find that the agreement with modeling is vague. Perhaps it would be useful to focus on these results.]

Re: We started the current study in an attempt to elucidate the effect of local field inhomogeneity on T_2 relaxivity of magnetic clusters. First, we confirmed that local field inhomogeneity can be artificially enhanced by introducing size discrepancy of IO NPs (i.e., IO-5 and IO-15) into assembled clusters (i.e., C3), which is responsible for the elevated T_2 relaxivity. These results further encouraged us to study the role of shape discrepancy of IO NPs (i.e., cube and plates) in

local field inhomogeneity and T_2 relaxivity of assembled clusters (i.e., C6 and C7), which was inspired by the fact that anisotropic shaped single IO NPs with larger effective radius exhibit larger r_2 values over their related spherical ones (refs. 14, 15, and 19, *Nat. Comm.* 4, 2266 (2013); *Nano Lett.* 12, 3127-3131 (2012); *ACS Nano* 8, 7976-7985 (2014)). The novelty in the current study is that we generalized the T_2 relaxivity to the local field inhomogeneity induced by IO clusters in terms of size and shape discrepancy within IO clusters. The overall study with different models together provide a general way to design high-performance T_2 contrast agents. Regarding the reviewer's concern about the modeling results, we have conducted two sets of simulation models to support models C6 and C7 (Supplementary Figs. 18 and 19, see also the response to Reviewer #3).

[It would be interesting to know how stable these NP clusters are in aqueous solution, as colloidal stability has been routinely demonstrated to be a problem in IONP systems.]

Re: We have tracked the colloidal stability of IO clusters C1-C3 through a period of 45 days by recording the hydrodynamic diameters using DLS measurement. The results indicate that the as-prepared IO clusters are stable in aqueous solution without obvious agglomerations and changes in hydrodynamic diameter, probably due to the PEG decorated surface on IO clusters. It is noteworthy that the obtained IO clusters can be stored with great stability in aqueous solution for at least three months. Accordingly, we have updated the data and discussed in the revised manuscript (Supplementary Fig. 3, Page 6).

Additional:

“It is noteworthy that the IO clusters are stable in aqueous solution for at least 45 days without obvious agglomerations and changes in hydrodynamic diameter (Supplementary Fig. 3), probably due to the PEG decorated surface on IO clusters.”

Supplementary Fig. 3 DLS measurements of the size changes of IO clusters C1-C3 in aqueous solution over a period of 45 days after preparation, indicating good colloidal stability.”

[There is no mention of how CNR is calculated in that the SNR calculation is unspecified. This begs the question of how reproducible the CNR results would be on another scanner, even of the same model and field strength.]

Re: The calculation of CNR and SNR was shown in the Methods part: $SNR = SI_{\text{mean}}/SD_{\text{noise}}$, $CNR = |SNR_{\text{tumor}} - SNR_{\text{liver}}|/ SNR_{\text{tumor}}$, where signal intensity (SI) was obtained in defined regions of interest (e.g., liver or tumor), standard deviation (SD) of background noise was measured in the phase-encoding region outside the imaging subject in order to account for any possible artifacts. Accordingly, we have modified the calculation details of CNR and SNR in the Methods part and in the main text.

The reviewer also brought up the issue of reproducibility of CNR on different scanners. This is due to the artifacts in MRI study which can be induced by both the equipment itself and the scanned object. First, the defective magnetic field and radiofrequency pulse would generate considerable phase inhomogeneity during MRI scan. Second, the scanned object, such as living mouse, may introduce artifacts due to motional and chemical shifts. In both cases, the reduced SNR and inaccurate CNR may lead to false-positive diagnosis in clinical practice. While the first case can be largely resolved by choosing specific acquisition sequence, such as T_2 rather than T_2^* sequence, or positioning the object as close to the magnetic center as possible, the second case can be partially altered by utilizing a respiratory gating device to reduce the artifacts caused by respiration. In our study, we tried our best to carefully combine these solutions to achieve optimal MRI acquisitions, so that we believe these results are considered reproducible in other scanners under similar set-ups.

[I feel that a different journal would be a better place to present this work, with much more detail provided on the LLG modeling, which should be expanded to a much larger number of particles to better represent the actual clusters being studied.]

Re: We are thankful to the reviewer's constructive comments on our work. In light of the responses above, we have circumspectly considered and carefully addressed the major concerns about the simplistic LLG models raised by this reviewer and reviewer #1. By applying multiple-particle assembly models, we show that the local field inhomogeneity induced by multiple particles imply obvious differences in field asymmetry around IO clusters C1, C2 and C3 in a more comprehensive manner than the 2-nanoparticle models (Supplementary Fig. 12). Moreover, we also provided a movie to view in 3D the field inhomogeneity induced by assembled cluster models C1, C2 and C3 (Supplementary Movie 1). Together with the revisions in response to the other reviewer's comments, we believe that the revised manuscript has been substantially improved. Therefore, we stand by our original belief that this study provides a fresh viewpoint of local field inhomogeneity to interpret T_2 shortening effect of MNPs generalized from single particles to assembled clusters and should be of great interest to the broad readership of Nature Communications.

Reviewers' comments:

Reviewer #2 (Remarks to the Author):

By answering to all the questions of the different reviewers, the authors have really improved their paper and provided supplementary informations confirming their novelty concerning the influence of the clustering on local magnetic field heterogeneity. This paper deserves to be published in Nature communications.

Reviewer #4 (Remarks to the Author):

[Minor revisions needed]

It seems that many of my and other reviewers' comments have been addressed in the present revision. There are still a few things that need to be addressed.

1) In response to Reviewer #1, the concept of field inhomogeneity most certainly was not introduced to describe the intrinsic defect of the magnet itself on MRI machines, resulting in T2*. This statement also does not distinguish between gross field inhomogeneity in B0 vs. the local field inhomogeneities generated by the multi-domain particles.

2) Reference 37 cites the *first* name of the author, rather than his last name (Carroll). Please fix this.

3) Line 66: "established *in* single-domain MNPs"

4) In the LLG section of Methods, there are no units attributed to γ . Please fix this. Also, you might consider using SI units, e.g., correct erg/cm to J/m.

5) With regards to the LLG simulations, is there any evidence that the clustering of the particles occurs in such an organized fashion? I think the modeling is better with this revision, but it would be nice to match up the distribution of particles with what is seen in the clustered TEM images.

6) The CNR calculation still leaves something to be desired. It is important to realize that the ROI for noise taken from the background will not yield true noise, due to the vendor-specific post-processing that occurs prior to seeing the MR image on a workstation. Rather, one should take a subtraction image in order to see what the real noise is, i.e., acquire two images with the same sequence parameters, and subtract them from one another to remove the "image signal", leaving just the noise. If possible, I suggest addressing this in such images are available to determine the noise. This approach should yield more consistent results across vendors with regards to calculating an absolute CNR. As it is, your CNR is going to be adversely affected by the SNR, which will be scanner-specific and post-processing-specific in ways that are not obvious.

The following are our point-by-point response (Re) to the Reviewer's comments (in Italics). Changes in the manuscript are marked in red.

Reviewer #2 (Remarks to the Author):

[By answering to all the questions of the different reviewers, the authors have really improved their paper and provided supplementary informations confirming their novelty concerning the influence of the clustering on local magnetic field heterogeneity. This paper deserves to be published in Nature communications.]

Re: We appreciate the reviewer's endorsement of our work.

Reviewer #4 (Remarks to the Author):

[Minor revisions needed]

[It seems that many of my and other reviewers' comments have been addressed in the present revision. There are still a few things that need to be addressed.]

Re: We greatly appreciate the reviewer's positive comments. Regarding the reviewer's additional concerns, we have made appropriate changes in the manuscript accordingly.

[1) In response to Reviewer #1, the concept of field inhomogeneity most certainly was not introduced to describe the intrinsic defect of the magnet itself on MRI machines, resulting in T_2^ . This statement also does not distinguish between gross field inhomogeneity in B_0 vs. the local field inhomogeneities generated by the multi-domain particles.]*

Re: Thanks for the reviewer's criticism of the concept of field inhomogeneity and we agree with the reviewer's point-of-view. In fact, we used 'local field inhomogeneity' induced by magnetic nanoparticles to distinguish between 'intrinsic field inhomogeneity' existing in MRI magnet B_0 . The 'intrinsic field inhomogeneity' existing in MRI magnet B_0 causes T_2^* relaxation. In practice, one can acquire T_2 using spin echo sequence to refocus the spin loss by T_2^* decay, which can theoretically diminish the participation of T_2^* relaxation. In our work, we mainly focused on the 'local field inhomogeneity' induced by magnetic nanoclusters, thus we acquired T_2 rather than T_2^* throughout our work. To avoid potential misleading, we have thoroughly checked the manuscript and revised these terms accordingly.

*[2) Reference 37 cites the *first* name of the author, rather than his last name (Carroll). Please fix this.]*

Re: We have revised the citation accordingly.

[3] Line 66: "established **in** single-domain MNPs".]

Re: We have revised in the manuscript accordingly.

[4] In the LLG section of Methods, there are no units attributed to γ . Please fix this. Also, you might consider using SI units, e.g., correct erg/cm to J/m.]

Re: We have added the unit of " $s^{-1}T^{-1}$ " for γ . The exchange stiffness constant "1.0e-6 erg/cm" is revised as "1.0e-11 J/m" (Page 20).

[5] With regards to the LLG simulations, is there any evidence that the clustering of the particles occurs in such an organized fashion? I think the modeling is better with this revision, but it would be nice to match up the distribution of particles with what is seen in the clustered TEM images.]

Re: The particles within IO clusters are randomly distributed based on TEM images. Indeed, the organization of particles within self-assembled clusters is technically challenging, which often requires elaborated design of surface modification and procedure optimization (refs. 48-49; *Science* **345**, 1149-1153 (2014); *Science* **338**, 358-363 (2012)). In our study, we intended to report a general mechanistic study of the local magnetic field inhomogeneity induced by artificially involved MNPs of different sizes and shapes. The LLG simulations were carried out on well-organized models due to the ease of computational processing and calculation. More importantly, the well-organized models, especially in the case of multi-particle models, provide an intuitive comparison of the local magnetic field inhomogeneity between them. We agree with the reviewer that in most cases it would be nice to match up the observation and simulation. However, additional complexity of the LLG simulating models would drastically burden the computational processing and calculation, which is limited by our current capability. Moreover, in the specific purpose of better comparing different models, we would like to preserve the original settings of our LLG simulating models. To explain this, we have made additional discussion in the manuscript accordingly (Page 15).

[6] The CNR calculation still leaves something to be desired. It is important to realize that the ROI for noise taken from the background will not yield true noise, due to the vendor-specific post-processing that occurs prior to seeing the MR image on a workstation. Rather, one should take a subtraction image in order to see what the real noise is, i.e., acquire two images with the same sequence parameters, and subtract them from one another to remove the "image signal",

leaving just the noise. If possible, I suggest addressing this is such images are available to determine the noise. This approach should yield more consistent results across vendors with regards to calculating an absolute CNR. As it is, your CNR is going to be adversely affected by the SNR, which will be scanner-specific and post-processing-specific in ways that are not obvious.]

Re: We fully agree with the reviewer's comments. Regarding the calculation accuracy of CNR, it is subjected to the calculation of SNR. According to the National Electrical Manufacturers Association (NEMA) standards publication (MS 6-2008, R2014), the subtracting method is a primary standard method to determine SNR in diagnostic MR imaging. This method requires to execute two scans sequentially of the same slice within less than 5 min elapsed time from the end of the first scan to the beginning of the second. In addition, the NEMA standards also provide an alternative single-image measurement procedure for SNR, which furnishes our calculating methods for SNR and CNR in this study. With no doubt, the subtracting method could yield more consistent CNR results. In our study, we acquired one-scan MR images of pre-injection and 1 h post-injection of contrast agents for each mouse. We thus referred to the single-image measurement procedure to calculate the SNR and CNR in this work. Further investigation of the CNR calculation and quantification across different MRI scanners by using IO cluster contrast agents is necessary. To explain the current calculating methods, we have made revisions in the manuscript (Methods) accordingly (Page 21).

[Other minor revisions]

Minor revisions/additions in the Data availability, References, Acknowledgements, and Figures and legends according to the manuscript checklist are marked in red.

REVIEWERS' COMMENTS:

Reviewer #4 (Remarks to the Author):

I feel that you have adequately addressed my remaining comments. While I would like to have better methodology for SNR/CNR calculations, I feel that you are clearly aware of the limitations of your single image method. Additionally, you have invoked the NEMA standard. In the future, I hope that you are able to obtain two images per the dual-image NEMA standard for SNR/CNR.

Recommend: Publish as-is

The following are our point-by-point response (Re) to the Reviewer's comments (in Italics).

Reviewer #4 (Remarks to the Author):

[I feel that you have adequately addressed my remaining comments. While I would like to have better methodology for SNR/CNR calculations, I feel that you are clearly aware of the limitations of your single image method. Additionally, you have invoked the NEMA standard. In the future, I hope that you are able to obtain two images per the dual-image NEMA standard for SNR/CNR.]

[Recommend: Publish as-is]

Re: Thanks for the reviewer's comments. In the future, we will conduct the dual-image protocols as described in the NEMA standard to execute more accurate SNR/CNR calculations. Again, we would like to express our deep appreciation to the reviewer's constructive comments on this work.